

# OSL rock surface exposure dating as a novel approach for reconstructing transport histories of coastal boulders over decadal to centennial timescales

Dominik Brill[1]*, Simon Matthias May[1], Nadia Mhammdi[2], Georgina King[3], Christoph Burow[1], Dennis Wolf[1], Anja Zander[1], Benjamin Lehmann[4], Helmut Brückner[1]

[1] Institute of Geography, University of Cologne, Köln, Germany
[2] Institut Scientifique, Laboratory LGRN and GEOPAC Research Center, Université Mohammed V, Rabat, Rabat, Morocco
[3] Institute for Geology, University of Lausanne, Switzerland
[4] Centro de Estudios Avanzados en Zonas Áridas (CEAZA), ULS-Campus Andrés Bello, Raúl Britán 1305, La Serena, Chile

*Correspondence to*: Dominik Brill (brilld@uni-koeln.de)

**Abstract**. Wave-transported boulders represent important records of storm and tsunami impact over geological timescales. Their use for hazard assessment requires chronological information that in many cases cannot be achieved by established dating approaches. To fill this gap, this study investigated, for the first time, the potential of optically stimulated luminescence rock surface exposure dating (OSL-RSED) for estimating transport ages of wave-emplaced coastal boulders. The approach was applied to calcarenite clasts at the Rabat coast, Morocco. Calibration of the OSL-RSED model was based on samples with rock surfaces exposed to sunlight for ~2 years, and OSL exposure ages were evaluated against age control deduced from satellite images. Our results show that the dating precision is limited for all boulders due to the local source rock lithology which has low amounts of quartz and feldspar. The dating accuracy may be affected by erosion rates on boulder surfaces of 0.06-0.2 mm/year. Nevertheless, we propose a robust relative chronology for boulders that are not affected by significant post-depositional erosion and that share surface angles of inclination with the calibration samples. The relative chronology indicates that (i) most boulders were moved by storm waves; (ii) these storms lifted boulders with masses of up to ~40 t; and (iii) the role of storms for the formation of boulder deposits along the Rabat coast is much more significant than previously assumed. Although OSL-RSED cannot provide reliable absolute exposure ages for the coastal boulders in this study, the approach has large potential for boulder deposits composed of rocks with larger amounts of quartz or feldspar, older formation histories and less susceptibility to erosion.

## 1. Introduction

Coastal boulders with masses of up to tens or hundreds of tons, located well above high tide level or far inland from the shoreline, are impressive evidence for the occurrence and impact of tsunamis and extreme storms (e.g. Engel and May, 2012; May et al., 2015; Cox et al., 2019). Such geological imprints may be preserved over periods that significantly exceed instrumental and historical records (Yu et al., 2009; Ramalho et al., 2015), making them valuable archives for long-term hazard assessment. Compared to sandy tsunami and storm deposits, which are used more commonly for this purpose, wave-transported boulders are abundant along rocky coastlines and can be preserved over geological time scales even in settings dominated by erosion (Paris et al., 2011). Furthermore, boulder transport may provide information on the magnitude of prehistoric tsunamis and storms that cannot be deduced from sandy sediments (Nandasena et al., 2011).





For coastal boulders to be valuable for hazard assessment, they have to provide information on the frequency of
the associated flooding events, which in turn requires chronological information on boulder displacement. Since
boulders unlike sandy storm and tsunami deposits typically lack a stratigraphic context, dating approaches rely on
chronometers related to the boulder rock itself or on constructive features attached to the boulder, such as marine
organisms or flow stones. Established dating approaches are based on radiocarbon ($^{14}$C) and U-series ($^{230}$Th/$^{234}$U)
dating of organic carbonates (e.g. Zhao et al., 2009; Araoka et al., 2013), and thus require coral boulders or the
presence of attached marine organisms, as well as coincidence between the death of these organisms and the
transportation event. Direct ages for the transport of coastal boulders were achieved by using terrestrial cosmogenic
nuclide surface exposure dating (Ramalho et al., 2015; Rixhon et al., 2017) and palaeomagnetic dating (Sato et al.,
2014), but due to intrinsic methodological limitations these approaches are restricted to certain boulder lithologies
and time scales.

The recently developed optically stimulated luminescence (OSL) rock surface dating technique (see review by
King et al., 2019) offers completely new opportunities for directly dating the transport of coastal boulders. While
the application of the more routinely used OSL rock surface burial dating technique (e.g. Simms et al., 2011;
Sohbati et al., 2015; Jenkins et al., 2018; Rades et al., 2018) is typically impeded for coastal boulders due to
logistical problems with sampling the (inaccessible) light-shielded bottom surfaces of clasts weighing several tons,
the OSL rock surface exposure dating (OSL-RSED) technique introduced by Sohbati et al. (2011) can be applied
to the light-exposed top surfaces of such clasts. For boulders that were overturned during wave transport and that
experienced negligible erosion and shielding of their top surfaces after deposition, post-transport exposure periods
may be estimated based on the time-dependent progression of OSL signal resetting, the so-called bleaching front,
into the uppermost millimetres to centimetres of the rock (Sohbati et al., 2012; Freiesleben et al., 2015; Lehmann
et al., 2018; Gliganic et al., 2019). OSL-RSED could therefore provide ages for coastal boulders that are not datable
by any other technique. OSL-RSED is applicable to a wide spectrum of lithologies, as long as they contain quartz
and/or feldspar, and to timescales of decades, centuries up to a few millennia.

Here, we present the first application of OSL-RSED to reconstruct storm and/or tsunami frequency patterns from
wave-emplaced boulders. All analyses were conducted on carbonatic sandstone boulders from the Atlantic coast
of Morocco, south of Rabat, that were previously documented by Mhammdi et al. (2008) and Medina et al. (2011).
Primarily, this study aims at evaluating the novel OSL-RSED technique for coastal boulders, which was achieved
by using artificially exposed rock surfaces for calibration of the bleaching model and by testing its performance
against age control deduced from satellite images and eyewitness accounts. The successfully validated model was
then applied to boulders of unknown age. While some of these boulders had previously been tentatively attributed
to the 1755 Lisbon Tsunami (Mhammdi et al., 2008; Medina et al., 2011), they lack robust chronological data.
Besides discussing limitations of the dating approach due to local OSL signal properties and erosion of post-
transport exposed boulder surfaces, we also discuss the future potential of this method and the implications of the
new OSL-RSED boulder ages for the long-term storm and tsunami hazard at the Atlantic coast of Morocco.

## 2. The OSL rock surface exposure dating model applied to coastal boulders

Conventional OSL dating relies on the accumulation of an energy dose (palaeodose) due to the impact of ionising
radiation over time (dose rate) on sand or silt grains shielded from sunlight. The palaeodose is proportional to the
burial age of the sediment and can be quantified by measuring the light emission (OSL signal) of quartz or feldspar





grains during stimulation with laboratory light. In natural settings, resetting of OSL signals takes place by sunlight exposure during sediment transport, so that sediment grains can provide information about the time that passed

since the last sunlight exposure (burial age).

The uppermost millimetres to centimetres of rock surfaces exposed to sunlight experience bleaching and accumulation of OSL signals at the same time. However, OSL signal resetting or bleaching is by far the dominant process in rocks with low environmental dose rates and Holocene exposure histories (Sohbati et al., 2012). For coastal boulders with dose rates of less than 1 Gy/ka and ages post-dating the stabilization of Holocene eustatic

sea level around its present position about six millennia ago (e.g. Khan et al., 2015), as investigated in this study, OSL signal accumulation can be neglected. The time-dependent evolution of OSL signals in the upper layer of exposed boulder surfaces can therefore be reduced to the term for OSL signal resetting, which following Sohbati et al. (2012) is expressed by

$$L(\mathrm{x}) = L_0 e^{-\overline{\sigma\varphi_0}t_e e^{-\mu x}},\tag{1}$$

where $L_0$ is the initial OSL signal intensity prior to exposure, L the remaining OSL signal at depth x (mm) after exposure, $t_e$ (s) the exposure time, $\overline{\sigma\varphi_0}$ (s$^{-1}$) the effective bleaching rate of the OSL signal at the rock surface (i.e. the product of the photo-ionisation cross section $\sigma$, and the light flux at the rock surface $\varphi_0$), and $\mu$ (mm$^{-1}$) the light attenuation coefficient of the rock.

Figure 1 illustrates how Equation (1) can be used to estimate the transport age of boulders overturned by waves.

When attached to the cliff, only the (usually bio-eroded) upper surface of a typical boulder in the pre-transport position is exposed to sunlight and experiences OSL signal resetting (Fig. 1a). Its shielded bottom side is only exposed to ionising radiation from radioactive elements in the surrounding rock and cosmic rays that, after a prolonged time, cause OSL signals to be in or close to field saturation (Fig. 1a). When overturned during transport, the new upper surface of the boulder in the post-transport position is suddenly exposed to sunlight and the

bleaching front starts to move into the rock (Fig. 1b); the same is true for the surfaces of quarrying niches that are formed by boulder detachment (Fig. 1b). In both cases, the exposure time can be estimated by fitting Equation (1) to the depth-dependent OSL signals measured in rock samples collected from these surfaces. The shielded bottom side of the boulder in the post-transport position is generally suitable for rock surface burial dating, by making use of the time-dependent dose accumulation in the previously bleached surface; due to inaccessibility of shielded

surfaces for sample collection this was not tried in this study.

## 3. Study area

### 3.1 Marine flooding hazard along the Atlantic coast of Morocco

The approximately 3000 km-long Moroccan Atlantic coast is exposed to swell waves, north Atlantic winter storms and rare tsunamis that cause erosion and/or flooding of low-lying areas. The energy of swell waves is strongest

along the central section of the Moroccan coast, between Agadir and Rabat, since it is not sheltered by the Canary Islands or the Iberian Peninsula; waves approach from the northwest to west and are significantly stronger during winter (Medina et al., 2011). The influence of Atlantic hurricanes is comparatively small (Fig. A1a) with only two former tropical storms recorded to have made landfall as tropical depressions (core pressure 988-1000 hPa) at the coast of Morocco and the southern Iberian Peninsula between 1851 and 2016 (Fig. 2a). Instead, maximum wave

heights are associated with winter storms that typically cross France or the UK (Fig. A1b), but may have tracks as



far south as Morocco (Fig. 2a). During recent winter storms within the last century, wave heights of up to 7 m (compared to regular swell heights of 0.5-1.5 m) have been observed at the Rabat coast (Mhammdi et al., 2020), associated with flooding of back-beach areas and waves overtopping the coastal cliff (Fig. A2).

An additional flooding hazard emanates from tsunamis triggered by earthquakes offshore of Portugal, between the

Azores triple junction and the Strait of Gibraltar, where the African and Eurasian plates converge at a rate of ~4 mm per year (Zitellini et al., 1999). After earthquakes in 1941, 1969 and 1975, Moroccan tide gauges recorded moderate tsunamis with waves <1 m. Further earthquakes, likely accompanied by tsunamis with impact in Morocco, are listed in historical catalogues (e.g. in 382 CE and 881 CE), but unambiguous reports of flooding only exist for the 1st November 1755 Lisbon Tsunami (Kaabouben et al., 2009). Triggered by a $M_w$ 8.5 earthquake,

probably due to the rare event of a combined rupture of different seismic structures (Baptista et al., 2003), the associated tsunami is the only known destructive flooding event at the Moroccan coast. Historical sources from Rabat describe the inundation of streets as far as 2 km inland, wreckage of ships in the harbour, and drowned people and camels (Blanc, 2009). Although numerical models indicate that the wave heights of 15 m mentioned in historical reports from Tanger and Safi are most likely exaggerated and that values of 2.5-5.0 m are more realistic

(Fig. 2a; Blanc, 2009; Renou et al., 2011), the effects of the 1755 tsunami on the coastal landscape of Morocco were nevertheless significant (e.g. Ramalho et al., 2018).

### 3.2 Exploiting geological evidence for hazard assessment – The Rabat boulder fields

While instrumental and historical records demonstrate the flooding hazard at the Moroccan coast due to both storms and tsunamis, all documented events except the 1755 Lisbon Tsunami were restricted to the last decades.

This does not allow for robust estimates of long-term tsunami and storm occurrence or of all possible magnitudes of storm surges and tsunami inundation. Most published regional geological tsunami and storm evidence for the pre-instrumental era is restricted to Spain and Portugal (e.g. Dawson et al., 1995; Hindson and Andrade, 1999; Lario et al., 2011; Costa et al., 2011; Feist et al., 2019), but fields of wave-emplaced boulders offer records of past storms and/or tsunamis for Morocco (Mhammdi et al., 2008; Medina et al., 2011) that could inform about the

regional long-term hazard if robust chronological data were available.

The most prominent boulder fields are reported from a 30 km long NE-SW oriented coastal section between Rabat and Skhirat (Fig. 2a,b), consisting of hundreds of boulders with estimated masses between a few and more than 100 t (Mhammdi et al., 2008; Medina et al., 2011). The geomorphology and geology of this area is characterised by a succession of coast-parallel, Pleistocene calcarenite ridges that are related to sea-level highstands and rest on

a Palaeozoic basement (Chakroun et al., 2017). A typical cross section (Fig. 2c,d) is composed of: (i) the intertidal platform with an active coastal cliff; (ii) the youngest lithified calcarenite ridge, formed during MIS 5; (iii) an inter-ridge depression, called Oulja, which may be flooded at high tide (the spring tide range is 2-3 m), and which is covered by recent and/or Holocene beach deposits; and (iv) an older calcarenite ridge, probably formed during MIS 7, including an inactive cliff (Medina et al., 2011; Chakroun et al., 2017; Chahid et al., 2017). Towards Rabat,

the younger calcarenite ridge is replaced by a simple sandstone platform (Fig. 2e).

As described by Mhammdi et al. (2008) and Medina et al. (2011), most of the calcarenite boulders were sourced from the active cliff (Fig. 2c). Since detachment is guided by lithological boundaries between the calcarenite and interbedded clay units, most of the boulders have platy shapes; only occasionally were boulders derived from subtidal positions and lifted up to 5 m vertically to the top of the first calcarenite ridge, as indicated by vermetids,

or sourced from younger sandstones covering the Oulja. The boulders are deposited as single clasts, clusters, or





imbricated stacks that rest on top or at the backward slope of the first calcarenite ridge, in the Oulja, or rarely at the seaward slope of the older calcarenite ridge up to 300 m inland (Fig. 2c). The position and orientation of bio-erosive rock pools formed on the surface of the youngest ridge (i.e. the pre-transport surface of most boulders) offers insights into transport modes. While some boulders moved by sliding only, others were overturned during

transport as indicated by down-facing rock pools on the pre-transport surface (Mhammdi et al., 2008; Medina et al., 2011). For some of the larger boulders, sliding movement by storm waves after their initial detachment from the cliff is documented on satellite images (Fig. A3). Movement of smaller boulders with up to 1 m³ (~2.5 t) was frequently observed after recent winter storms such as Hercules/Christina in January 2014 (Mhammdi et al., 2020). At some places along the coast between Rabat and Casablanca even boulders exceeding 10 t have been pushed

landward during recent winter storms (Mhammdi et al., 2020).

## 4. Methods

Boulders sampled for dating were characterized in the field with regard to their position, orientation, dimension and surface taphonomy. Distance from the active cliff and elevation above mean high tide level were measured using a laser range finder. Boulder volume estimates (V) are based on tape measurements of a- (length), b- (width)

and c-axes (height) and an empirical correction factor of 0.5 (Engel and May, 2012) using

$$V = (a * b * c) * 0.5, \tag{2}$$

To calculate boulder weights, volumes are multiplied with boulder densities ($\rho_B$) determined individually for each sample using the Archimedean principle of buoyancy in water following

$$\rho_B = \rho_W * \frac{w_a}{w_a - w_w}, \tag{3}$$

with $w_a$ = weight of the sample in air, $w_w$ = weight of the sample in water and $\rho_W$ = density of sea water (1.02 g/cm³). Surface orientation and inclination of sampled boulders were measured with a compass.

For OSL-RSED, samples of approximately 10 cm³ were collected from selected boulder surfaces using a combination of a battery-driven rock drill, hammer and chisel. Rock samples were wrapped in black plastic bags and brought to the Cologne Luminescence Laboratory (CLL) for further processing under dimmed red-light

conditions. First, a circular rock saw was used to cut ~5 cm thick surface slabs, from which cores of ~1 cm diameter and ~4 cm length were extracted using a bench drill (Proxxon Professional) with water cooled diamond core bits. After immersion in resin (Crystalbond 509, the resin was tested to have no OSL emission) and subsequent oven drying to stabilize fragile parts of the sandstone cores, they were cut into ~0.7 mm thick slices using a water-cooled low speed diamond saw (Bühler Isomet 1000) with 0.3 mm blade thickness. Slices were gently crushed with a

mortar to obtain polymineralic sand grains that were fixed on aluminium cups using silicon grease in monolayer. Separation of pure quartz and/or potassium feldspar for the grains of each slice, standard practice in conventional OSL dating, was not feasible due to the large number of slices and the small amount of polymineralic grains per slice.

To optimize the information extracted from the polymineralic samples, all luminescence measurements followed

a post-IRSL-BSL protocol (e.g. Banerjee et al., 2001) that records an infrared stimulated luminescence (IRSL) signal at 50 °C for 160 s, followed by a blue stimulated luminescence (BSL) signal at 125 °C for 40 s (Tab. A2). Measurements were performed on a Risø TL/OSL DA20 reader equipped with an U340 filter for signal detection.





All thermal treatments were performed with heating rates of 2 °C/s. In the post-IRSL-BSL protocol, stimulation with infrared LEDs specifically bleached luminescence signals originating from feldspar (feldspar IRSL). This

reduced the contribution of feldspar signals to the BSL signal of quartz (quartz BSL), which unlike feldspar is insensitive to infrared stimulation (cf. Bailey, 2010).

For validation of the post-IRSL-BSL protocol, pure quartz and potassium feldspar extracts in the 150-200 µm grain-size fraction were prepared for the light-shielded parts (i.e. >5 cm below surface) of the 10 cm³ sample blocks of HAR 1-1 and TEM 3-1. Sample preparation followed standard coarse grain procedures including dry

sieving, treatment with 10% HCl and 10% $H_2O_2$, density separation (potassium feldspar<2.58 g/cm³<2.62 g/cm³<quartz<2.68 g/cm³), and 40% HF etching in the case of quartz. Dose recovery experiments with signal resetting in a solar simulator for 24 hours and administering of a ~12 Gy laboratory beta dose, as well as continuous wave fitting of quartz BSL components using the **R** package "Luminescence" version 0.9.0.88 (Kreutzer et al., 2019) were performed for both the pure quartz and the polymineralic fraction. Additional preheat-plateau tests and

palaeodose determinations were conducted on quartz extracts following a conventional SAR protocol according to Murray and Wintle (2003) (Tab. A3). Dose rates are based on high-resolution gamma spectrometry and the conversion factors of Guerin et al. (2011). Conventional OSL burial ages for the sandstone were calculated from burial doses and dose rates using the DRAC software version 1.2 (Durcan et al., 2015).

For OSL-RSED, the natural OSL signals ($L_n$) and the OSL signals in response to a ~12 Gy test dose ($T_n$) of the

post-IRSL-BSL protocol were measured for the polymineralic grains of all crushed slices to generate plots of OSL signal versus depth below the boulder surface. The depth-dependent $L_n/T_n$ data of each core (mean of two aliquots) were normalized to the core's individual background level calculated from the average of the deepest 5-10 slices. The normalized data of all cores of a sample were then averaged (arithmetic mean and standard error) to receive a mean signal-depth curve for each rock sample; only apparent outliers, i.e. cores with signal-depth trends completely different from all other cores of the sample, were excluded from averaging. The mean signal-depth

curves were fitted with Equation (1) using the unweighted rock surface exposure dating function in the R package "Luminescence" (Burow, 2019) and the software OriginPro (version 8.5). Shared µ values for each site and shared $\overline{\sigma\varphi_0}$ values for flat calibration surfaces were determined using the "global fit" function that allows the fitting of multiple signal-depth curves at the same time. Post-depositional erosion has recently been shown to exercise a

strong effect on the depth of the bleaching front, and thus the apparent age, of exposed rock surfaces (Sohbati et al., 2018; Lehmann et al., 2019a,b; Brown and Moon, 2019). Their potential effects were therefore modelled using the approach of Lehmann et al. (2019a).

## 5. Results

### 5.1 Boulders selected for OSL surface exposure dating

Samples for OSL-RSED were collected from nine boulders at four different sites along the Rabat coast in July 2016, including Rabat (RAB), Haroura (HAR), Temara (TEM) and Val d'Or (VAL) (Fig. 2b). Boulders selected for dating were composed of carbonate-cemented sandstone (calcarenite) with clear signs of overturning during transport, indicated by down-facing rock pools and/or fresh-looking post-transport surfaces (Fig. 3d). To ensure comparable preconditions for sunlight exposure, only surfaces without significant shielding by vegetation, other

boulders or water, and wherever possible without significant inclination of their top surfaces were sampled. Most sampled boulders, thus, rested in supratidal positions and had relatively smooth post-transport surfaces (RAB 1,



HAR 1, HAR 2, TEM 3, VAL 4, VAL 6). However, boulders from the intertidal platform with post-transport rock pools (VAL 1, Fig. 3h) or boulders with higher surface roughness probably due to increased sea spray influence (TEM 2 and RAB 5, Fig. 3g) were also sampled for assessing the effects of post-depositional erosion on dating accuracy. In addition, surfaces of niches in the active cliff, exposed after detachment of the associated boulders, were sampled at Haroura (HAR 3, Fig. 3e) and Temara (TEM 4).

The characteristic features of all sampled boulders – including post-transport position, arrangement, shape, dimension, orientation of the sampled surface and taphonomy of boulder surfaces – are summarized in Table 1. Satellite images covering the last 50 years (Google Earth images from 2001 to 2019, Corona images from 1966), field observations for very young features, and, in case of VAL 1, the depth of post-depositional rock pools helped to roughly constrain when the boulders and niches were deposited or formed (see Tab. A1 for a summary). Precise age control by observations of local residents confirmed the movement of boulder TEM 3 during winter storm Hercules/Christina in January 2014 ($t_e$ = 2.5 years), and the formation of niche TEM 4 (sampled in September 2018) between the 2016 and 2018 field surveys, most likely during the unnamed winter storm in February 2017 ($t_e$ = 1.5 years) (Fig. A4). Corona satellite images provide minimum ages of 50 years for boulders RAB 1, RAB 5, VAL 1, VAL 4 and HAR 2, since all of them were identified at their present position on images from April 1966 (Fig. A5, A6, A7, A8, A9). However, considering the up to 45 cm deep post-depositional rock pools on the surface of VAL 1 and assuming typical rates of bio-erosion in the range of up to 1 mm/year (Kelletat, 2013), boulder VAL 1 is probably much older than 50 years, at least a few centuries. All other boulders and niches could not be identified on the 1966 satellite images due to their limited resolution. However, these clasts did not change their position between 2001/2004 and 2019 (Fig. A9, A10, A11), equalling minimum ages of 12-15 years (Tab. A1).

### 5.2 Luminescence properties of the dated sandstone

Comparative measurements on polymineralic grains and potassium feldspar extracts on sample HAR 1-1 show that post-IRSL-BSL signals from the polymineralic aliquots of all four sites are (i) the dominant emission compared to IRSL signals, and (ii) relatively unaffected by a feldspar signal contribution (Fig. A12). Therefore, OSL-RSED in this study was based on the mainly quartz derived post-IRSL-BSL signal of polymineralic aliquots. Experiments on pure quartz extracts of sample HAR 1-1 revealed adequate OSL properties in terms of rapidly decaying signals dominated by the fast component (Fig. A12a,b), independence of thermal treatment for the selected preheat temperature (Fig. A13), and good reproducibility of laboratory doses (dose recovery ratios of 1.02-1.08). Similarly, suitable OSL properties, i.e. signals dominated by the quartz fast component (Fig. A12c) and successful dose recovery experiments, are also documented for post-IRSL-BSL signals of polymineralic aliquots.

When plotted against their depth below the boulder surface, test dose corrected and normalized mean post-IRSL-BSL signals from the uppermost 15 mm of each sample (note that signal-depth curves of each sample are based on 2 to 5 cores with 2 aliquots per slice) showed a general increase from completely reset signals at the rock surface towards a constant background level deeper in the rock (Fig. 4, Fig. A14, A15). The background levels reflected a quartz palaeodose of ~40-50 Gy or an age of ~80-100 ka (measured on HAR 1-1 and TEM 3-1, Tab. A4), which is below the sample-specific saturation level of 50-120 Gy. The robustness of the average post-IRSL-BSL-depth trends used for dating is supported by good reproducibility of signals derived from different aliquots of the same slice (Fig. 4a), and reasonable correlation of different cores from the same sample (Fig. 4b, Fig. A14, S15). Where





signal-to-noise ratios also allowed feldspar IRSL signals to be analysed (i.e. at TEM and RAB), these showed bleaching fronts that intruded deeper into the rock compared to the post-IRSL-BSL signal (Fig. 4c, Fig. A11).

**5.3 Calibration of the OSL rock surface exposure dating model using artificially exposed surfaces**




To estimate boulder ages with OSL-RSED, measured post-IRSL-BSL signal-depth data were fitted with the bleaching model described in Equation (1). Besides the exposure time ($t_e$), the bleaching model contains two further *a priori* unknown parameters: the effective OSL signal bleaching rate at the rock surface ($\overline{\sigma\varphi_0}$), and the light attenuation in the rock ($\mu$). These vary with geographical location and rock type, respectively, and have to be determined individually for each location and lithology prior to dating. Since determination on the basis of first order principles was not successful in earlier studies (Sohbati et al., 2012), for the Rabat site these parameters were obtained empirically by fitting Equation (1) to calibration samples with known exposure ages (e.g. Sohbati et al., 2012; Lehmann et al., 2018).

For this, fresh rock surfaces were exposed during the first field survey in July 2016 and sampled during the second survey in September 2018, equivalent to an exposure time of ~2.15 years. A total of five calibration samples, at least one rock sample from each site, were collected to account for potential site-to-site variability (CAL samples in Tab. 1). Exposures were created directly on the top surfaces of boulders RAB 5, TEM 3 and VAL 4 (Fig. A17b,d), as well as by placing previously unexposed rock samples collected from boulders HAR 1 and VAL 4 on the roof top of a nearby house (Fig. A17a,c). Since the effective luminescence decay rate ($\overline{\sigma\varphi_0}$) is sensitive to the inclination and orientation of the dated rock surfaces (Gliganic et al., 2019), all exposure surfaces except from TEM 3 CAL, which had the same inclination as the associated dating sample, were orientated approximately horizontally.



In a first step, local values for $\overline{\sigma\varphi_0}$ were determined. Since all samples in this study were collected within a radius of less than 20 km, the local light flux should be similar for all surfaces with comparable inclination and orientation. This was supported by fitting each calibration sample individually (Fig. 5, Tab. 2), reflecting systematic differences of $\overline{\sigma\varphi_0}$ only between the inclined calibration surface of TEM 3-1 CAL ($1.23 \times 10^{-5}$ s$^{-1}$) and the horizontal calibration surfaces of all other calibration samples ($2.7 \times 10^{-7}$ to $4.7 \times 10^{-8}$ s$^{-1}$). We therefore determined a shared $\overline{\sigma\varphi_0}$ value for all horizontal surfaces by simultaneously fitting the respective calibration samples, using $\overline{\sigma\varphi_0}$ (shared) and $\mu$ (individual best-fit value for each sample) as free variables and an exposure age of 2.15 years as a fixed parameter (Tab. 2). This resulted in shared $\overline{\sigma\varphi_0}$ values of $1.2(\pm2.3) \times 10^{-5}$ s$^{-1}$ for the inclined surface and $9.2(\pm2.0) \times 10^{-8}$ s$^{-1}$ for the horizontal surfaces.




The second step was the estimation of local values for $\mu$. Light attenuation in the rock may be influenced by small-scale variations in lithology and therefore $\mu$ should have boulder-dependent values (Gliganic et al., 2019). However, fitting Equation (1) to the post-IRSL-BSL signal-depth data of individual samples revealed $\mu$ values with huge uncertainties (Fig. 5a-5e, Tab. 2). Sample-dependent best-fit $\mu$ values ranged between 0.26 and 3.5 mm$^{-1}$ for the boulder samples in this study (Tab. 2), while literature values for the BSL signal of quartz sandstone and quartzite are in the range of 0.9-1.3 mm$^{-1}$ (cf. Sohbati et al., 2012, Gliganic et al., 2019). This indicates that sample-specific values may not only be imprecise but, due to large measurement uncertainties, may also be inadequate for some samples. Since the estimation of shared $\mu$ values for several rock samples can improve the accuracy of the estimate significantly (Lehmann et al., 2018), the use of shared $\mu$ values for all boulders from an individual site (i.e. RAB, HAR, TEM and VAL) was chosen as a reasonable and necessary compromise. The assumption of very similar $\mu$ values for all boulders from one individual site is supported by their very similar lithology, since all of



the boulders are derived from the same local calcarenite facies. Site-dependent μ values were obtained by fitting Equation (1) to all samples from a site at the same time, using μ (shared) and exposure time (individual for each sample) as free variables, and the shared $\overline{\sigma\varphi_0}$ value for horizontal surfaces determined in the previous step as a fixed parameter (only for TEM 3-1, TEM 4-1 and TEM 3-1 CAL the $\overline{\sigma\varphi_0}$ value for inclined surfaces was used).

Site-averaged μ values vary between 1.04±0.26 mm⁻¹ at RAB and 1.54±0.31 mm⁻¹ at TEM (Tab. 2), which seem to be much more realistic when compared to the literature values for BSL signal attenuation in quartz sandstone and quartzite of 0.9-1.3 mm⁻¹ (cf. Sohbati et al., 2012, Gliganic et al., 2019).

**5.4 Model validation and dating of boulders with unknown transport history**

OSL exposure ages for all non-calibration boulder and niche samples were derived by fitting their post-IRSL-BSL
signal-depth profiles with Equation (1) using the site-averaged μ values and the shared $\overline{\sigma\varphi_0}$ value for horizontal surfaces (the value for inclined surfaces was only used for TEM 3-1 and TEM 4-1) as fixed parameters (Tab. 2). Complete incorporation of both μ and $\overline{\sigma\varphi_0}$ uncertainties resulted in relatively large fitting uncertainties (Fig. 6a) that were finally reflected in the error margins of the OSL surface exposure ages. The fitted post-IRSL-BSL signal-depth curves of all dating samples and the associated exposure ages are summarized in Figure 6b and Table 2,
respectively. To evaluate the accuracy of model-derived exposure ages, they were compared with minimum transport ages deduced from satellite images, eyewitness observations and the depth of bio-erosive rock pools (Fig. 6c). The OSL surface exposure ages of most samples agree with the control ages, i.e. ages either post-dated the minimum age or showed overlap within their dating uncertainties. However, the exposure ages of samples RAB 1-2, VAL 1-1, VAL 1-2, HAR 1-1 and HAR 2-1 were too young, i.e. they pre-dated the minimum control ages.

**5.5 Modelling post-depositional erosion of boulder surfaces**

In order to explore whether erosion offers a plausible explanation of the age underestimations recorded for samples RAB 1-2, HAR 1-1, HAR 1-2, VAL 1-1 and VAL 1-2, the potential effect of erosion on the luminescence bleaching profiles was modelled using the analytical approach of Lehmann et al. (2019a). The modelled sample ages ($t_{exp\ mean}$) and minimum independent ages ($t_{age\ control}$) were used as model inputs, together with the shared values
of μ and $\overline{\sigma\varphi_0}$  (Tab. 2). 50 different erosion rates from 0.001 mm/year to 1 mm/year were tested together with 50 different times for the onset of erosion ($t_s$) ranging from 1 year to the independent sample age (both variables were sampled equidistantly in log space). The misfits between modelled and measured values were determined and paths with normalised misfit >0.99 were retained. The sensitivity of the calculated erosion rates to the independent age was also evaluated by contrasting the results calculated for sample VAL 1-2 for independent ages of 50 years,
450 years and 6000 years, which reflect the minimum exposure age based on satellite images, a plausible estimate of the boulder turning age based on the depth of post-depositional rock pools and finally the time when Holocene sea level reached approximately its present position. The calculated erosion rates vary dependent on $t_s$ (Fig. 7), thus to facilitate comparison, erosion rates for $t_s$ of ten years are contrasted between samples (Tab. A5). The modelled erosion rates increase with increasing surface age, from 0.05 mm/year assuming an age of 50 years, to
0.20 mm/year assuming an age of 450 years, and to 0.40 mm/year assuming an age of 6000 years. Thus, erosion rate estimates based on minimum ages should be regarded as minimum values. Minimum erosion rates varied from 0.13 mm/year (RAB 1-2) to <0.01 mm/year (VAL 1-1), maximum values (based on maximum ages) reach 0.32 to 0.40 mm/year (VAL 1-1 and VAL 1-2).



## 6. Discussion

### 6.1 Performance of OSL surface exposure dating on coastal sandstone boulders

The OSL surface exposure ages derived for boulders and niches from the Rabat coast show two striking characteristics: (1) All exposure ages are associated with relatively large dating uncertainties compared to previous applications of OSL-RSED (e.g. Sohbati et al., 2012; Lehmann et al., 2018); and (2) five of the 13 dated boulder samples yield OSL exposure ages that underestimate minimum ages deduced from satellite imagery and rock-pool depth, even when their uncertainties are considered (Fig. 6c).

The low dating precision achieved in this study is mainly the result of the boulder source rock, a late Pleistocene calcarenite. All rock samples dated in this study display strongly scattered post-IRSL-BSL signal-depth data (e.g. Fig. 4 and Fig. 5) that entail large fitting uncertainties, imprecisely constrained $\mu$ and $\overline{\sigma\varphi_0}$ parameters and, eventually, large dating uncertainties. OSL signal scatter is primarily due to dim post-IRSL-BSL signals with not more than a few hundred photon counts in the analysed signal interval. Since pure quartz extracts of the same samples proved to be rather sensitive (Fig. A12), dim post-IRSL-BSL signals must be the result of low percentages of quartz on the carbonate-rich polymineralic aliquots used for dating. Additional signal scatter is introduced by spatial variations of the post-IRSL-BSL signal accumulated prior to exposure ($L_0$). Since post-IRSL-BSL signals in the relatively young source rocks of the boulders (i.e. 40-50 Gy and 80-100 ka) are not in field saturation, they depend on mineralogy-induced dose rate differences within the rock. Thirdly, a small contamination of post-IRSL-BSL signals by feldspar emissions remains in all dated samples. If the amount of feldspar varies from aliquot to aliquot, varying contributions of feldspar emissions to the post-IRSL-BSL signals from polymineralic aliquots will introduce additional scatter. While OSL exposure ages of rocks with more suitable luminescence properties are also affected by fitting uncertainties due to mineralogical heterogeneities (Meyer et al, 2018) and core-to-core variations of OSL signal resetting (Sellwood et al., 2019), previous studies demonstrated that lithologies with brighter quartz signals in polymineralic samples (e.g. quartzite or quartz-dominated sandstone) or stronger feldspar signals to avoid using quartz OSL for dating (e.g. granite or gneiss) can provide much higher dating precision than achieved for the Rabat boulders (Sohbati et al., 2012; Freiesleben et al., 2015; Lehmann et al., 2018; Gliganic et al., 2019).

Although large post-IRSL-BSL signal scatter may also affect dating accuracy, since it prevents using individual $\mu$ values for each sample as suggested e.g. by Gliganic et al. (2019), the unambiguous disagreement between exposure ages and age control for five of the boulder samples (Fig. 6c) is interpreted to result from inadequate $\overline{\sigma\varphi_0}$ values and post-depositional erosion. In the constrained geographical area visited in this study, $\overline{\sigma\varphi_0}$ should be comparable for all boulder surfaces as long as they share the same aspect and inclination (e.g. Sohbati et al., 2018). However, if calibration and dating samples do not share surface inclination and aspect, the use of a shared $\overline{\sigma\varphi_0}$ value is inappropriate, as observed in controlled bleaching experiments (Gliganic et al., 2019) and indicated by the systematic differences of $\overline{\sigma\varphi_0}$ between calibration samples with inclined and flat surfaces in this study. The clearly too young OSL exposure ages of samples HAR 2-1 and RAB 1-2, i.e. 25±8 and 11±3 years, although these boulders were overturned at least 50 years ago (Fig. 6c), could both reflect the mismatch between their inclined surfaces and fitting with a $\overline{\sigma\varphi_0}$ that was determined on flat calibration surfaces. Future boulder dating studies should ensure calibration samples with comparable inclination and orientation to the dating samples.

Besides inadequate model calibration, OSL rock surface exposure ages become inaccurate when their OSL signal-depth curves are affected by environmental factors beyond the exposure time. Since OSL-RSED is restricted to



the uppermost few mm or cm of rock surfaces, the position and shape of the bleaching front is very susceptible to erosion (Sohbati et al., 2018; Lehmann et al., 2019a,b; Brown and Moon, 2019). For soft sandstone boulders in the coastal zone as dated here, the combination of sea-spray and rain-induced weathering and strong winds is likely to cause erosion of grains at the exposed post-transport surfaces (e.g. Mottershead, 1989). By yielding erosion rates from 0.06 to 0.20 mm/year, inversion of the rock surface-exposure data for boulder samples that clearly underestimate age control (i.e. RAB 1-2, HAR 1-1, HAR 1-2, VAL 1-1 and VAL 1-2) supports the assumption of

significant erosion for some of the boulders dated in this study (Fig. 7).

Such impact of erosion agrees with expectations based on geomorphological evidence for boulders with post-transport surfaces covered by bio-erosive rock pools, such as boulder VAL 1. Since the lower part of this boulders is lying in the intertidal zone, it is regularly covered by sea spray and overtopping waves. Surfaces between bio-erosive rock pools, which can form with erosion rates of up to 1 mm/year (e.g. Kelletat, 2013), were sampled in

the case of VAL 1-1 and VAL 1-2. For these samples relatively large modelled erosion rates of 0.20 mm/year, when assuming an age of ~450 years based on rock-pool depth and bio-erosion rates of 1 mm/year (Fig. 7a), may therefore be realistic. These data illustrate the spatial heterogeneity in erosion rates for some of the coastal boulders sampled and the importance of careful sample location selection. Erosion rates are assumed to be much lower for boulders in supratidal positions, as indicated by much smoother post-transport surfaces (see Tab. 1). This is

consistent with inverse modelling on boulder HAR 1, whose OSL exposure age of 9±1 years slightly underestimates the minimum age of 15 years (Fig. 6c). This indicates a comparatively low erosion rate of 0.06 mm/year, consistent with its flat and apparently smooth post-transport surface (Fig. 7b). Our data suggest that some influence of erosion cannot unambiguously be ruled out even for calcarenite boulders with apparently smooth surfaces, and all OSL-RSED ages for boulders in this study should be interpreted with caution.

Other environmental factors that might affect OSL exposure ages are assumed to be negligible for all dated boulders. The post-transport surfaces of all boulders are bare of vegetation and not shielded by topography or houses. The surfaces of boulders in the intertidal zone (i.e. VAL 1) may be overtopped by waves during stronger storms (particularly contemporaneous with high-tide conditions), but periods with submersion are insignificantly short compared to the total exposure time. Likewise, the exposure duration of the calibration surfaces, i.e. another

important parameter for model calibration, had no negative effect on dating accuracy. The exposure time of ~2 years used in this study was more than sufficient to generate pronounced bleaching fronts in all calcarenite samples. Although model calibration generally benefits from calibration samples with long, and in the best case several different, exposure durations, even shorter exposure intervals than 2 years would have sufficed. In boulder samples with bright IRSL signals, these were even better bleached than the associated post-IRSL-BSL signals (Fig. 4c),

potentially because longer wavelengths that feldspar signals are sensitive to are less attenuated by the rock than the shorter wavelengths (Ou et al., 2018) that bleach quartz signals (Wallinga, 2002). While IRSL signals were not used in this study due to insufficiently bright signals for most samples, the application of IRSL instead of post-IRSL-BSL signals may reduce the time required for calibration to durations as short as a few months (Freiesleben et al., 2015; Ou et al., 2018).

**6.2 New information on storm and tsunami hazard at the Atlantic coast of Morocco**

Knowing the chronology of boulder transport can help to better assess the local flooding hazard at the Rabat coast. Energetic waves during storms and tsunamis will generally exacerbate the effects of coastal flooding in the course of climate-induced sea-level rise (Nicholls et al., 2018). It is therefore of paramount interest whether coastal





inundation strong enough to lift boulders at the Rabat coast only occurred during the very rare tsunami events,

such as the 1755 Lisbon Tsunami, or also during much more frequent winter storms.

Comparison with satellite images showed that OSL-RSED ages are definitely inaccurate for boulders affected by severe post-depositional erosion (VAL 1-1 and VAL 1-2, squares in Fig. 6c) and for boulder samples with significantly inclined surfaces (HAR 2-1 and RAB 1-2, stars in Fig. 6c); the associated OSL exposure ages cannot be considered for any further interpretation. All other boulder samples, including those with apparently smooth

surfaces, were likely affected to some extent by erosion as well. Slight age underestimation, thus, cannot be excluded and their exposure ages should be interpreted carefully. We nevertheless are confident that the latter provide valuable relative chronological information for boulder transport that is shown in Figure 8a and allows differentiation between boulder ages.

The reliability of this relative chronology is supported by correlation between OSL exposure ages and the surface

taphonomy of the associated boulders and niches (Fig. 8a, b). Exposure ages younger than ~10 years were achieved for boulders and niches with smooth surfaces and fresh fractures, i.e. taphonomy classes 4 and 5 (TEM 4, TEM 3, HAR 1; Fig. 8b1). Boulders with exposure ages between ~10 and ~100 years are characterised by smooth surfaces with very scarce lichen or algae cover, i.e. taphonomy classes 3 and 4 (HAR 3, RAB 1, VAL 6, TEM 2; Fig. 8b2). Finally, boulders with exposure ages older than ~100 years are characterised by weathered fractures and rougher

surfaces, i.e. taphonomy classes 2 and 3 (VAL 4, RAB 5; Fig. 8b3,b4). According to the chronology presented here, with OSL exposure ages of 152±52 years (VAL 4) and 577±247 years (RAB 5) and rather rough/weathered rock surfaces, these boulders are the only clasts that may have been moved by the 1755 Lisbon Tsunami. However, with masses of 16-24 t and positions on the intertidal platform (RAB 5) or on top of cliffs 3-4 m above sea level, they do not systematically differ from the other dated boulders in terms of wave power required for transportation.

Although the relative chronology does not unambiguously allow for correlating individual boulders with specific historical storms or tsunamis, two important conclusions with regard to the local flooding hazard can be drawn from the dataset. Firstly, the relative chronology in Figure 8a implies that most boulders at the Rabat coast were detached from the cliff and overturned by storm waves. The large spread of OSL exposure ages between a few years and several centuries indicates that numerous transport events were responsible for the formation of the dated

boulders. Since the 1755 Lisbon Tsunami was the only tsunami with significant flooding at the Moroccan Atlantic coast during the last 1000 years (Kaabouben et al., 2009), boulder transport dominated by tsunamis is assumed to have resulted in more significant clustering of ages around ~260 years ago.

Secondly, correlation of exposure ages and masses of the associated boulders shows that storm waves were capable of lifting much larger boulders than observed during recent winter storms. At the Rabat coast, observations from

the last decade are restricted to the lifting of smaller boulders (Mhammdi et al., 2020), while boulders larger than ~5 t were only observed to move by sliding (Fig. A3). However, boulders with OSL exposure ages that clearly postdate the 1755 Lisbon Tsunami and therefore must have been lifted by storms reach up to 38 t (RAB 1). These storm boulders yield comparable or even larger masses than boulders that, based on their exposure ages, might have been transported and overturned during the 1755 Lisbon Tsunami (i.e. VAL 4 and RAB 5 with masses of 16-

24 t). Of course, we cannot exclude that the largest boulders at the Rabat coast, such as VAL 1 with ~65 t that could not be dated with OSL-RSED due to strong erosion of their post-transport surface, can exclusively be overturned by tsunamis. Nevertheless, in agreement with hydrodynamic experiments (Cox et al., 2019) and observations after recent tropical cyclones (e.g. May et al., 2015), our results support the perception that storm waves significantly contribute to boulder quarrying along cliffs and may be considered an important driver for the


Earth **Surface**
**Dynamics**
Discussions

evolution of wave-emplaced coarse-clast deposits worldwide, including boulders with masses of several tens of
tons that have previously been associated with tsunamis. It is, therefore, likely that also other boulders documented
along the Atlantic coasts of Morocco (Mhammdi et al., 2008; Medina et al., 2011), and the Iberian Peninsula
(Whelan and Kelletat, 2005; Scheffers and Kelletat, 2005; Costa et al., 2011), which have tentatively been related
to the 1755 Lisbon Tsunami and potential predecessors previously but mainly lack sound chronological data, in
fact represent storm boulders.

## 7. Conclusions

OSL rock surface exposure dating was for the first time applied to coastal boulders overturned during wave
transport. Successful calibration of the bleaching model using surfaces exposed for ~2 years and evaluation of
OSL exposure ages against satellite images indicate the potential of the approach for boulders with limited post-
depositional erosion and with surface inclination in agreement with that of the calibration samples. Although fitting
uncertainties as a consequence of low amounts of quartz and potassium feldspar in the source rock introduced
relatively large dating uncertainties, and although a bias due to post-depositional erosion cannot be excluded even
for boulders with smooth surfaces, OSL rock surface exposure dating provides a relative chronology for boulders
that could not be dated with any other approach so far. This relative chronology indicates a large variability of
boulder ages, most of them different from the only tsunami event at the Rabat coast within the last 2000 years.
Thus, OSL exposure ages suggest that even boulders weighing ~40 t were moved and overturned by storm waves.
This supports the conclusion of previous studies that storms rather than tsunamis can be the most important driver
for the formation of coastal boulder deposits in general.

While OSL-RSED offered important relative chronological information for the Rabat coastal boulders but could
not provide absolute ages, the approach offers a powerful tool for dating boulder deposits with more favourable
lithologies. Magmatic rocks, such as granites, are not only significantly less susceptible to erosion, typically they
also allow measurement of the luminescence signal of potassium feldspar. Different from the quartz signals of the
calcarenite used in this study, IRSL signals of potassium feldspar measured on polymineralic aliquots do not suffer
from contamination by other minerals and are typically much brighter than those of quartz. Such lithological
properties promise to reduce the uncertainties and inaccuracies related to OSL surface exposure dating of coastal
boulders in this study significantly.

### Author contributions

D.B. and S.M. conceived the study and obtained funding. D.B., S.M., N.M. and G.K. conducted fieldwork. N.M.
provided site information and eyewitness accounts. D.B., S.M., D.W. and A.Z. established the equipment for
sample preparation at the CLL. D.B. conducted the measurements and evaluated the data. C.B. provided the R-
scripts for data analyses and helped with data fitting. S.M, G.K and B.L. helped with analysing and interpreting
the data. H.B. along with all other authors commented on the manuscript at all stages.

### Data availability

The complete dataset of $L_x/T_x$ values of all samples that were used for the fitting of luminescence depth data in
this study will be made available on PANGEA. It can be made available for reviewers if required.





**Competing interests**

"The authors declare that they have no conflict of interest."

**Acknowledgements**

This study was funded by the Deutsche Forschungsgemeinschaft (DFG, German Research Foundation, GZ: BR
5023/3-1 and MA 5768/2-1). The authors are grateful to M. Malika Ait and A. Chiguer for their support during
the first field survey in Morocco in July 2016. We would also like to thank the Moroccan Association of
Geosciences in the person of its President, Prof. Medina Fida for its support.

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

**Figures and tables**

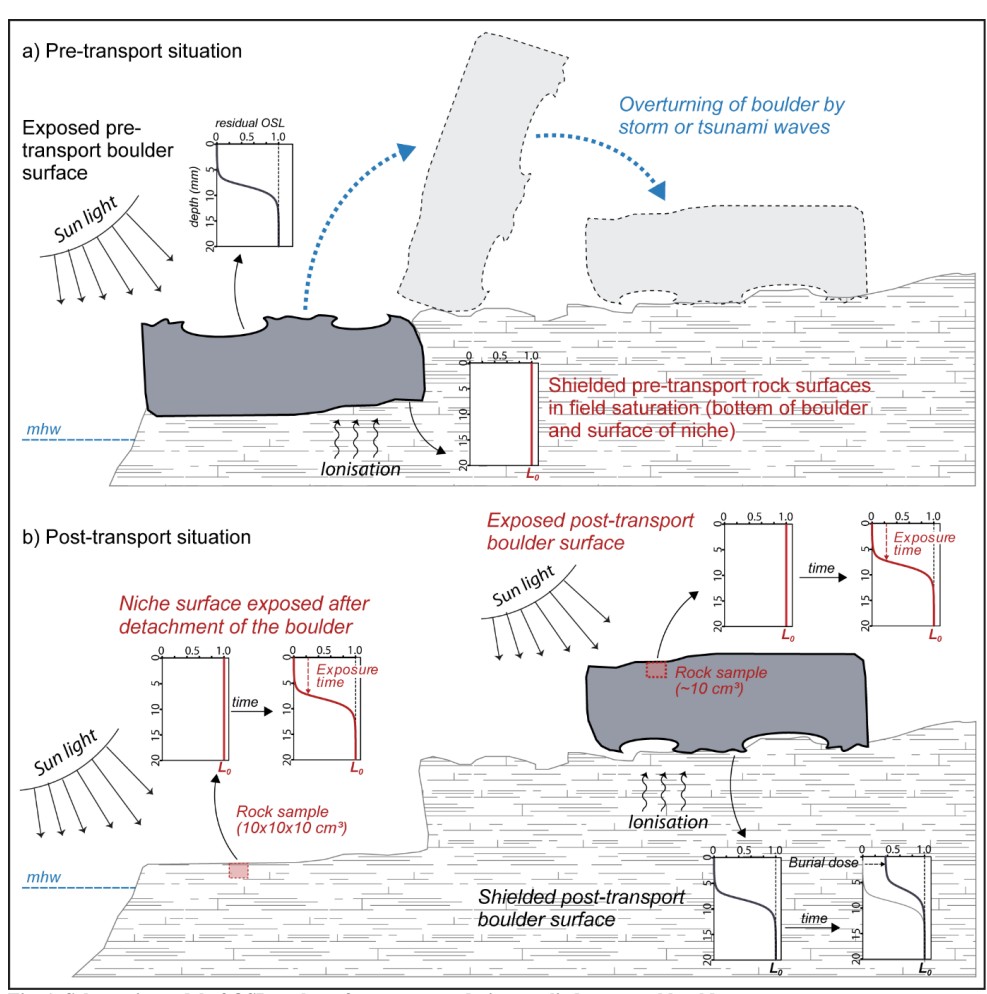


Fig. 1. Schematic model of OSL rock surface exposure dating applied to coastal boulders.



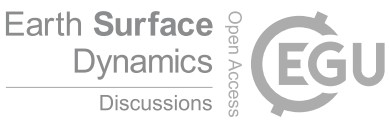

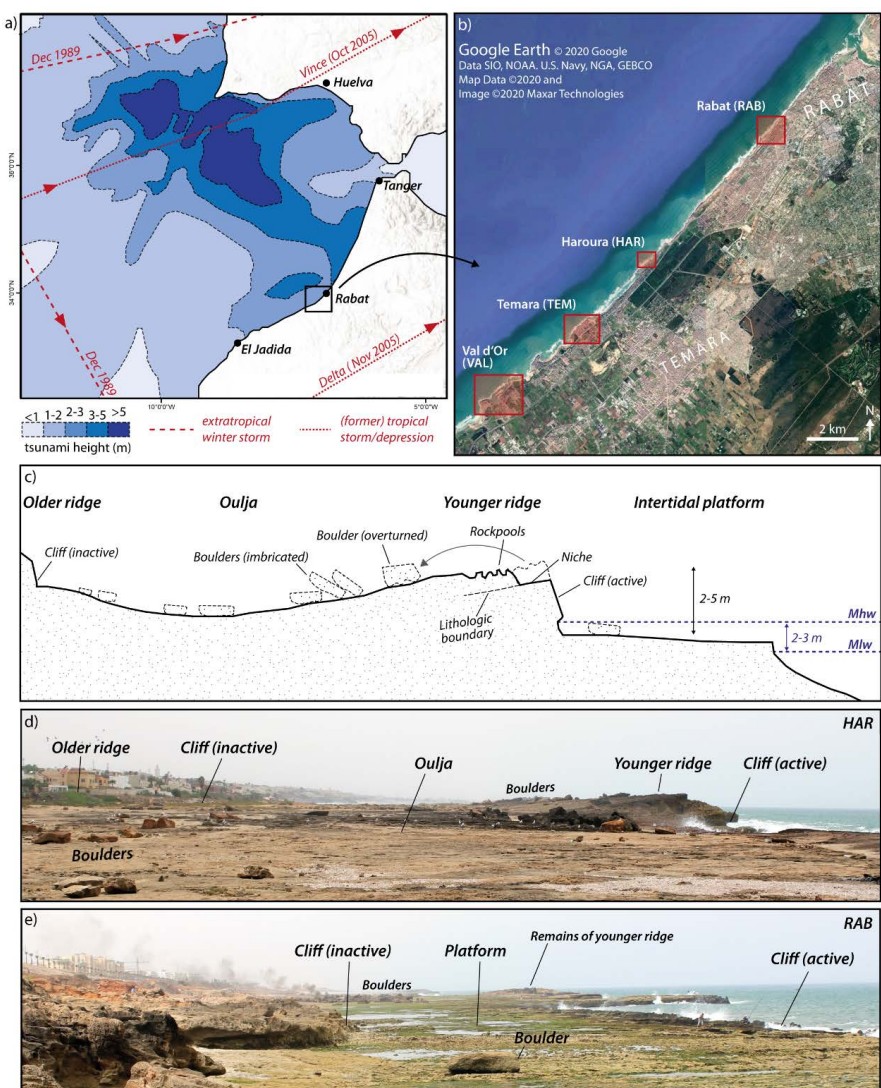

**Fig. 2: Flooding hazard and geomorphological setting of the Rabat coast. a) Exposure of the Moroccan Atlantic coast to tsunamis and storms, including modelled wave heights for the 1755 Lisbon tsunami (Renou et al., 2009), tracks of former tropical storms crossing the area between 1851 and 2016 (NOAA, 2019), and extratropical winter storms in the period 1989-2009 (Reading University, 2019). b) The Rabat coast with the four study sites (based on Google Earth images). c) Schematic geomorphological cross section through the Rabat coast at Haroura (HAR, modified from Mhammdi et al., 2008). d) The coastal platform at Haroura as shown in c) (view towards Southwest). e) The coastal platform at Rabat (RAB, view towards Southwest). ©**




Earth **Surface**
Dynamics
Discussions




**Fig. 3: Coastal boulders at the Rabat coast.** Satellite images of Val d'Or taken at low tide (a) and high tide (b) illustrate different boulder settings on top of the younger ridge, within the Oulja and on the intertidal platform (Google Earth images from July 2018 and February 2016). c) Boulder VAL 4 as part of a stack of imbricated boulders in ridge top position. d) Down-facing rock pools of the former cliff surface at the bottom surface of RAB 5. e) Niche HAR 3 formed

by detachment of the associated boulder. (f-h) Surface roughness of the sampled boulders varies from smooth (HAR 1), over slightly weathered (TEM 2), to rock-pool covered (VAL 1).

Earth **Surface**
**Dynamics**
Discussions

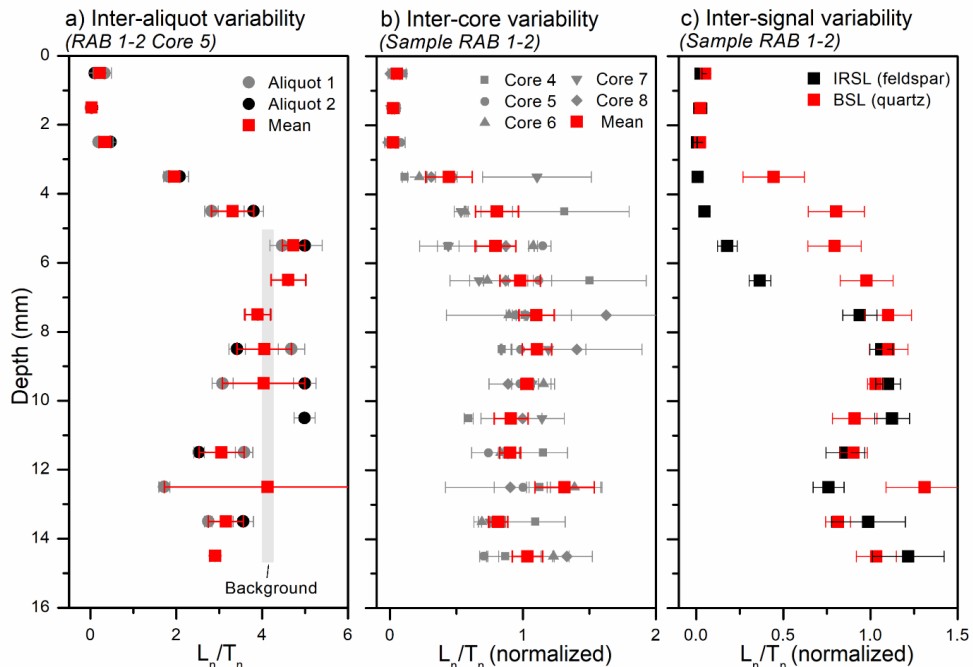

**Fig. 4: Exemplary OSL signal-depth data for boulders from the Rabat coast (RAB 1-2). a) Inter-aliquot variations of post-IRSL-BSL signals in core 5 of sample RAB 1-2. b) Variability of post-IRSL-BSL signals from different cores of the sample. c) Comparison of quartz post-IRSL-BSL and feldspar IRSL signals measured in the same post-IRSL-BSL protocol (mean values based on 5 cores each).**






**Fig. 5: Fitting of calibration samples. Individual fitting of the five calibration samples (a-e), and joint fitting of all calibration samples with horizontal surfaces using shared μ values for each site and a shared $\overline{\sigma\varphi_0}$ for all samples with flat surfaces (f). Fixed parameters are shown in black, calculated parameters in red (lower left corner of a-f).**






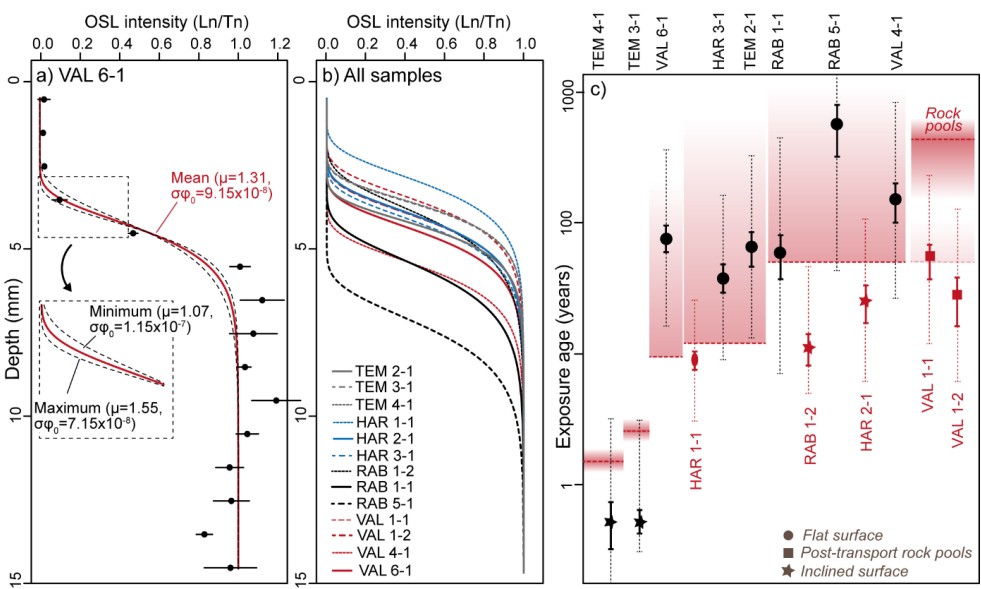

**Fig. 6: Fitting of post-IRSL-BSL signal-depth data and comparison of OSL exposure ages with age control. a) Fitting of sample VAL 6-1 with fixed μ and $\overline{\sigma\varphi_0}$. Fitting uncertainties due to the uncertainties of μ and $\overline{\sigma\varphi_0}$ are highlighted in the close up. b) Model fits for all dated samples (based on mean values for μ and $\overline{\sigma\varphi_0}$). c) Comparison of modelled exposure ages (symbols with error bars) and age control from satellite images, eyewitness observations and depth of post-transport rock pools (indicated by red shaded areas). Exposure ages in agreement with control ages are shown in black, those too young for the control ages in red.**



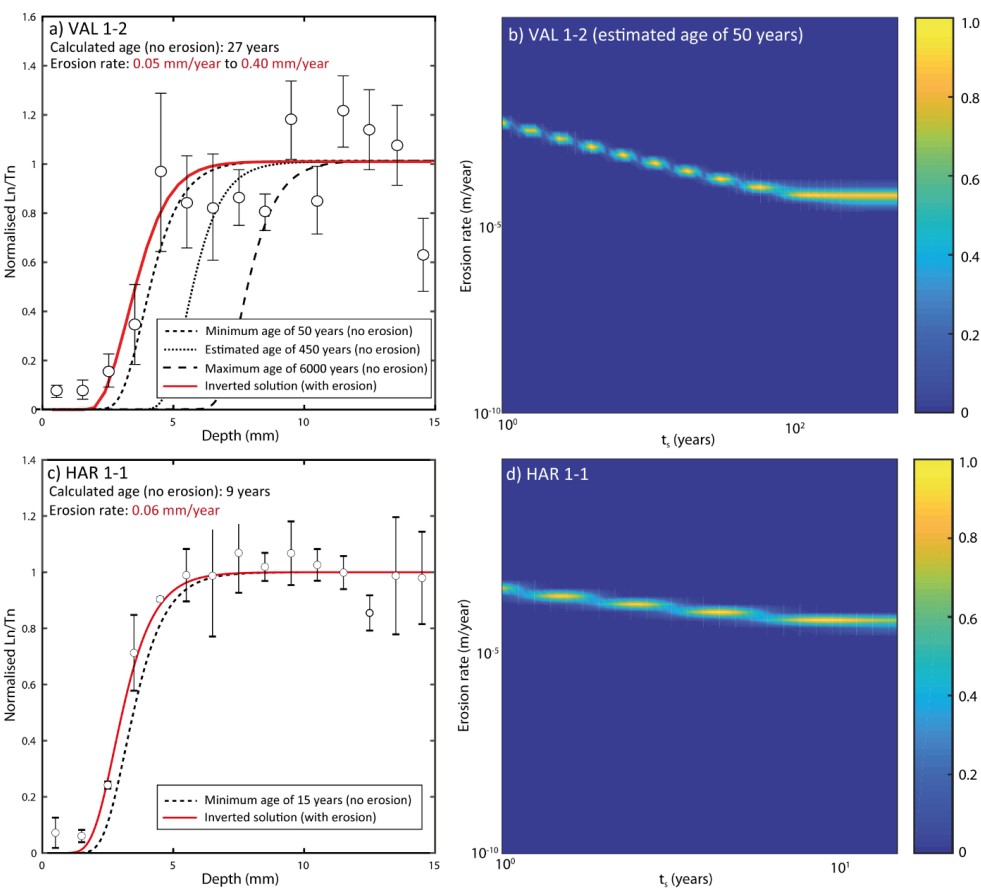

**Fig. 7: Inverse modelling of post-deposition erosion rates using the approach of Lehmann et al. (2019a). The effect of erosion on the OSL signal-depth profiles of samples VAL 1-2 (a, b) and HAR 1-1 (c, d) was evaluated using the shared** 710 **µ and $\overline{\sigma\varphi_0}$ from Table 2 as model input. In case of VAL 1-2 erosion rates were estimated for the minimum age (50 years), a realistic age estimate based on rock-pool depth (450 years), and the maximum age (6000 years). For HAR 1-1 only the minimum age of 15 years was used. Erosion rates are sensitive to changes of $t_s$ (b, d). The erosion rates reported in a) and b) are based on $t_s$ values of 10.**

Earth **Surface**
**Dynamics**
Discussions

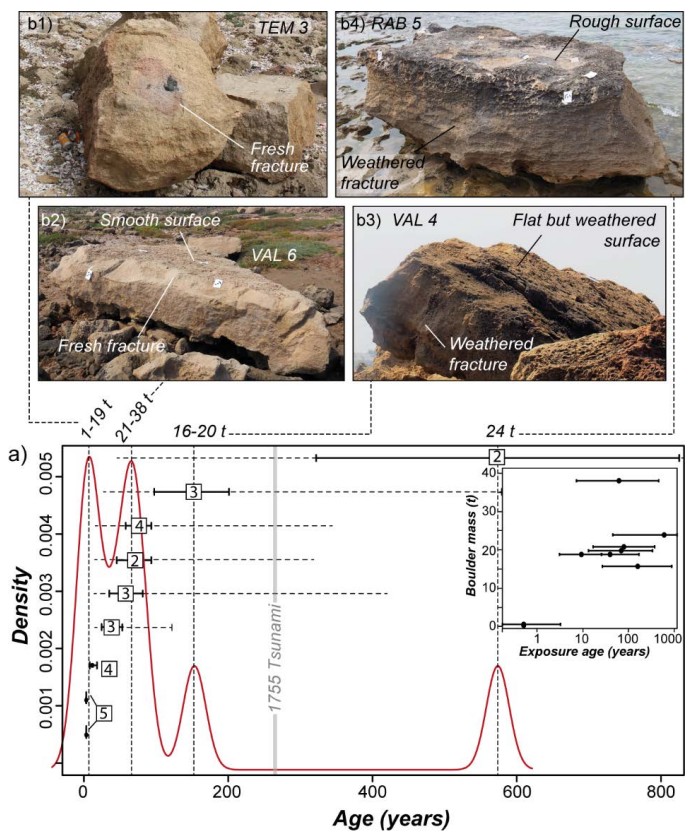

**Fig. 8: Relative chronology of boulder transport. a) Exposure ages of all boulders that do not clearly underestimate the control ages presented as KDE plot (dotted error bars with consideration of μ and $\overline{\sigma\varphi_0}$ uncertainties). The numbers in squares refer to the taphonomy classes described in the text and the caption of Table 1. Inset: Correlation between boulder mass and OSL rock surface exposure ages. b) Photographs documenting the taphonomy of boulders with different OSL rock surface exposure ages. Each photo is correlated with a KDE peak in a) and a boulder mass by dashed lines.**





| Boulder | Depositional setting | | | Boulder dimension | | | | Sampled surface | | Sample | |
|---|---|---|---|---|---|---|---|---|---|---|---|
| | Lat/Long (°) | Position | Group | Shape | Vol. (m³) | Density (g/m³) | Mass (t) | Orientation | Taphonomy | Dating | Calibration |
| RAB 1 | 34.007964 -6.869395 | Supratidal platform 2.7 m asl, 40 m from cliff | Single | Irregular | 15.3 | 2.45 | 37.5 | 0° 25° SE | 3 | RAB 1-1 RAB 1-2 | - |
| RAB 5 | 33.999778 -6.877759 | Intertidal platform (surface above tides) | Single | Cubic | 9.8 | 2.4 | 23.5 | 0° | 2 | RAB 5-1 | RAB 5-1 CAL |
| HAR 1 | 33.953904 -6.927609 | Oulja 2 m asl, 35 m from cliff | Single | Platy | 8.2 | 2.3 | 18.9 | 0° | 4 | HAR 1-1 | HAR 1-1 CAL** |
| HAR 2 | 33.953525 -6.928669 | Top of younger ridge 4.5 m asl, 16 m from cliff | Imbricated | Platy | 3.6 | 2.4 | 8.6 | 25° NW | 4 | HAR 2-1 | - |
| HAR 3* | 33.954182 -6.927901 | Niche in active cliff (intertidal) | - | Platy | - | - | - | 0° | 3 | HAR 3-1 | - |
| TEM 2 | 33.927066 -6.961859 | Top of younger ridge 4.2 m asl, 35 m from cliff | Single | Platy | 8.3 | 2.4 | 19.9 | 0° | 2 | TEM 2-1 | - |
| TEM 3 | 33.926842 -6.961915 | Top of younger ridge 4.5 m asl, 38 m from cliff | Single | Irregular | 0.3 | 2.4 | 0.7 | 25° W | 5 | TEM 3-1 | TEM 3-1 CAL |
| TEM 4* | 33.927949 -6.960674 | Niche in active cliff (supratidal) | - | Platy | - | - | - | 25° W | 5 | TEM 4-1 | - |
| VAL 1 | 33.909435 -6.989803 | Intertidal platform (surface above tides) | Single | Cubic | 26.2 | 2.25 | 59.0 | 10° SE | 1 | VAL 1-1 VAL 1-2 | - |
| VAL 4 | 33.907733 -6.991505 | Top of younger ridge 3 m asl, 25 m from cliff | Imbricated | Platy | 7.3 | 2.2 | 16.1 | 0° | 3 | VAL 4-1 | VAL 4-1 CAL1 VAL 4-1 CAL2** |
| VAL 6 | 33.906084 -6.993316 | Oulja 2.5 m asl, 80 m from cliff | Single | Platy | 8.9 | 2.3 | 20.5 | 0° | 4 | VAL 6-1 | - |

**Tab. 1: Characteristics of dated boulders. Lat/Long = Latitude/Longitude, * = niche at coastal cliff, ** = calibration sample on nearby roof top, taphonomy classes 1 to 5 with 1 – post-transport rock pools, 2 – rough post-transport surface covered with lichens/algae, 3 – smooth post-transport surface with scarce lichen/algae cover, 4 – smooth post-transport surface without/hardly any lichens/algae and fresh fractures, and 5 – fresh post-transport surfaces and fractures.**





| Sample | Cat. | Indiv. $\mu$ (mm$^{-1}$) | Shared $\mu$ (mm$^{-1}$) | Indiv. $\sigma\varphi_0$ (s$^{-1}$) | Shared $\sigma\varphi_0$ (s$^{-1}$) | $t_{exp}$ mean (yrs) | $t_{exp}$ Min-Max (yrs) | $t_{age\,control}$ (yrs) |
|---|---|---|---|---|---|---|---|---|
| RAB 5-1 CAL | Calibration | 0.73±0.35 | 1.04±0.26 (fix) | **4.2x10$^{-8}$±2.9x10$^{-8}$** | | 2.15 (fix) | - | 2.15 |
| HAR 1-1 CAL | | 1.31±0.55 | 1.19±0.23 (fix) | **2.7x10$^{-7}$±5.0x10$^{-7}$** | | 2.15 (fix) | - | 2.15 |
| VAL 4-1 CAL I | | 1.21±0.27 | 1.31±0.24 (fix) | **2.4x10$^{-7}$±1.3x10$^{-7}$** | | 2.15 (fix) | - | 2.15 |
| VAL 4-1 CAL II | | 1.08±0.34 | 1.31±0.24 (fix) | **5.9x10$^{-8}$±1.1x10$^{-8}$** | **9.2x10$^{-8}$±2.0x10$^{-8}$** (fix) | 2.15 (fix) | - | 2.15 |
| TEM 3-1 CAL | | 1.65±0.43 | 1.54±0.31 (fix) | **1.2x10$^{-5}$±2.3x10$^{-5}$** | **1.2x10$^{-5}$±2.3x10$^{-5}$** | 2.15 (fix) | - | 2.15 |
| RAB 1-1 | Dating | 2.93±4.81 | 1.04±0.26 (fix) | - | 9.2x10$^{-8}$±2.0x10$^{-8}$ (fix) | **59±22** | 7-441 | >50 |
| RAB 1-2 | | 1.17±0.39 | 1.04±0.26 (fix) | - | 9.2x10$^{-8}$±2.0x10$^{-8}$ (fix) | **11±3** | 5-46 | >50 |
| RAB 5-1 | | 0.26±0.06 | 1.04±0.26 (fix) | - | 9.2x10$^{-8}$±2.0x10$^{-8}$ (fix) | **577±247** | 43-3560 | >50 |
| HAR 1-1 | | 1.38±0.22 | 1.19±0.23 (fix) | - | 9.2x10$^{-8}$±2.0x10$^{-8}$ (fix) | **9±1** | 3-25 | >15 |
| HAR 2-1 | | 1.16±0.53 | 1.19±0.23 (fix) | - | 9.2x10$^{-8}$±2.0x10$^{-8}$ (fix) | **25±8** | 6-105 | >50 |
| HAR 3-1 | | 1.16±0.53 | 1.19±0.23 (fix) | - | 9.2x10$^{-8}$±2.0x10$^{-8}$ (fix) | **38±9** | 9-164 | >15 |
| TEM 2-1 | | 1.34±0.51 | 1.54±0.31 (fix) | - | 9.2x10$^{-8}$±2.0x10$^{-8}$ (fix) | **66±20** | 13-320 | >15 |
| TEM 3-1 | | 2.17±0.99 | 1.54±0.31 (fix) | - | 1.2x10$^{-5}$±2.3x10$^{-5}$ (fix) | **0.5±0.1** | 0.1-3.1 | ~2.5 |
| TEM 4-1 | | 0.78±0.23 | 1.54±0.31 (fix) | - | 1.2x10$^{-5}$±2.3x10$^{-5}$ (fix) | **0.5±0.2** | 0.1-3.3 | ~1.5 |
| VAL 1-1 | | 1.10±0.40 | 1.31±0.24 (fix) | - | 9.2x10$^{-8}$±2.0x10$^{-8}$ (fix) | **53±16** | 12-232 | >50 ~450 |
| VAL 1-2 | | 1.01±0.44 | 1.31±0.24 (fix) | - | 9.2x10$^{-8}$±2.0x10$^{-8}$ (fix) | **27±11** | 6-125 | >50 ~450 |
| VAL 4-1 | | 0.87±0.31 | 1.31±0.24 (fix) | - | 9.2x10$^{-8}$±2.0x10$^{-8}$ (fix) | **152±52** | 26-846 | >50 |
| VAL 6-1 | | 3.54±4.19 | 1.31±0.24 (fix) | - | 9.2x10$^{-8}$±2.0x10$^{-8}$ (fix) | **76±18** | 16-362 | >12 |

**Tab. 2: Summary of model parameters for all calibration and dating samples. Bold numbers indicate values that were calculated by the model. Cat. = sample category, Indiv. = individual, $t_{exp}$ mean = exposure ages based on fixed $\mu$ and $\overline{\sigma\varphi_0}$ values without their uncertainties, $t_{exp}$ Min-Max = exposure age range with consideration of $\mu$ and $\overline{\sigma\varphi_0}$ uncertainties.**

Earth **Surface**
Dynamics
Discussions

**Appendices**

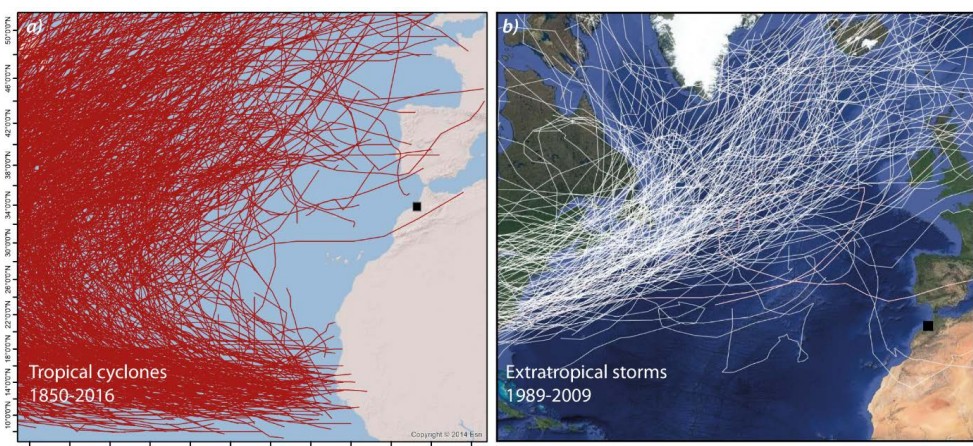

**Fig. A1: Storm hazard at the Rabat coast. a) Tracks of historical (1850-2016) tropical storms in the North Atlantic (NOAA, 2019); even aged tropical cyclones (tropical depressions) rarely strike the coastlines of the eastern Atlantic as far south as Spain or Morocco. b) Tracks of the 200 strongest extratropical storms in the North Atlantic 1989-2009 (Atlas of extratropical storms, University of Reading, 2019); most winter storms cross northern Europe, storm tracks as far south as Spain or Morocco are very rare.**

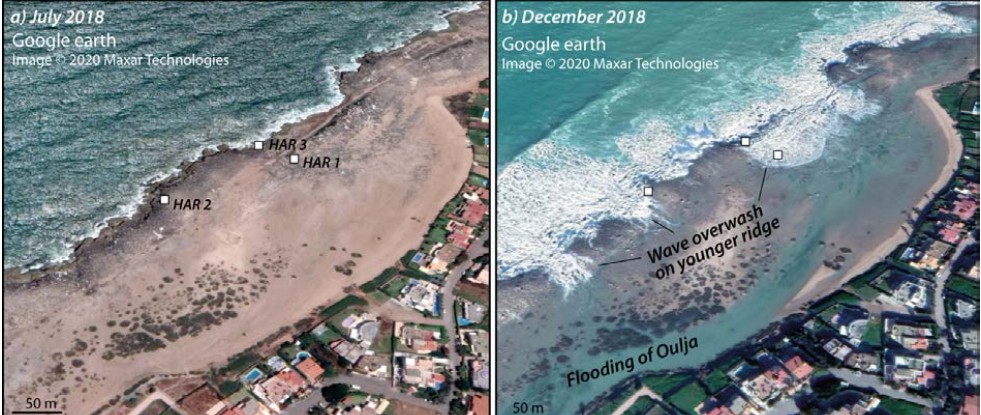

**Fig. A2: Storm waves and coastal flooding at Haroura. a) During normal wave conditions, all sampled boulders are located above tide level. b) Flooding of the Oulja and local wave overwash reaching up to 50 m landward of the shoreline during a winter storm in December 2018. Both scenes are based on Google Earth images.**



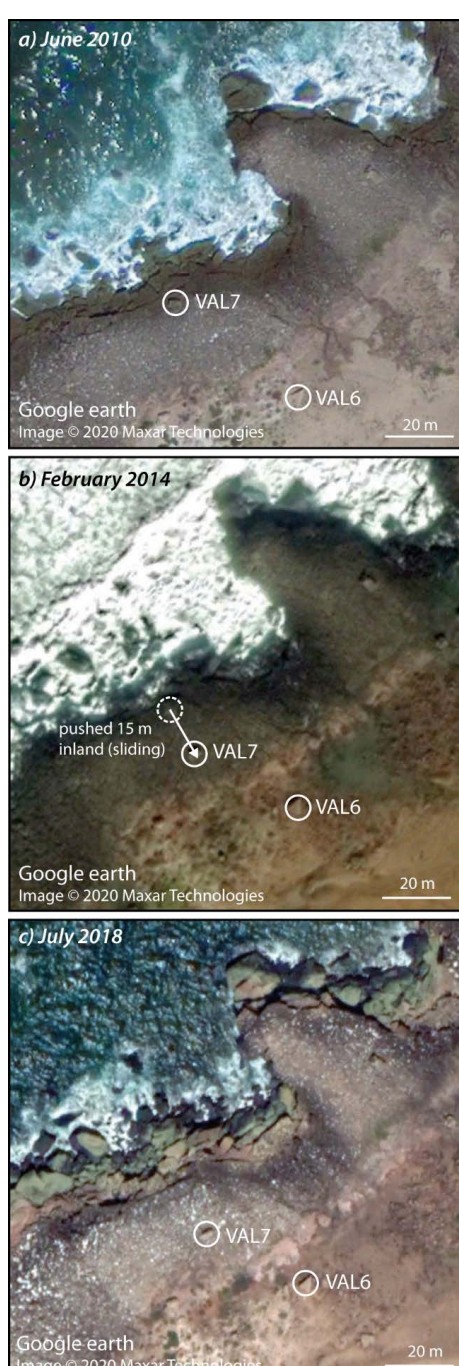

**Fig. A3: Storm transport of boulders recorded on satellite images (Google Earth). a) Positions of boulders VAL 6 and 7 in June 2010. b) Positions of the same boulders in February 2014. While VAL 6 remains stable, pushed by storm waves VAL 7 has moved for about 15 m perpendicular to the shoreline. c) After relocation in February 2014, both boulders remained in stable positions until July 2018. All scenes are based on Google Earth images.**



| Boulder/Niche | Corona 1966 | Google Earth | Observation | Age (years) |
|---|---|---|---|---|
| RAB 1 | at present position or slightly seaward but already overturned | at present pos. in 2001 | - | >50 |
| RAB 5 | at present position | at present pos. in 2001 | - | >50 |
| HAR 1 | - | at present pos. in 2001 | - | >15 |
| HAR 2 | at present position, a-axis slightly turned | at present pos. in 2001 | - | >50 |
| HAR 3 | - | at present pos. in 2001 | - | >15 |
| TEM 2 | - | at present pos. in 2001 | - | >15 |
| TEM 3 | - | - | deposited in Feb 2014 | ~2.5 |
| TEM 4 | - | - | formed between Jul 2016 and Sept 2018 | ~1.5 |
| VAL 1 | at present position | at present pos. in 2004 | Up to 45 cm deep post-transport rock pools | >50 ~450* |
| VAL 4 | at present position | at present pos. in 2004 | - | >50 |
| VAL 6 | - | at present pos. in 2004 | - | >12 |

**Tab. A1: Summary of age control for boulder movement and niche formation in the form of satellite images and own observations. pos. = position, - = no clear evidence. *Minimum age estimate based on the depth of post-depositional rock pools and empirical rates of bio-erosion in the order of 1 mm/year (Kelletat, 2013).**

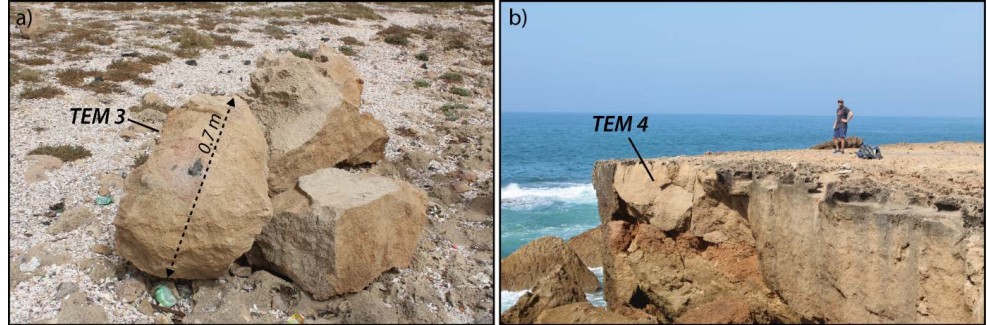

**Fig. A4: Boulder TEM 3 (a) was transported to its onshore location during winter storm Hercules/Christina in February 2014 as reported by local residents. Niche TEM 4 was formed between the field surveys in Juli 2016 and September 2018, most likely by a winter storm in 2017.**



Earth **Surface**
**Dynamics**
Discussions

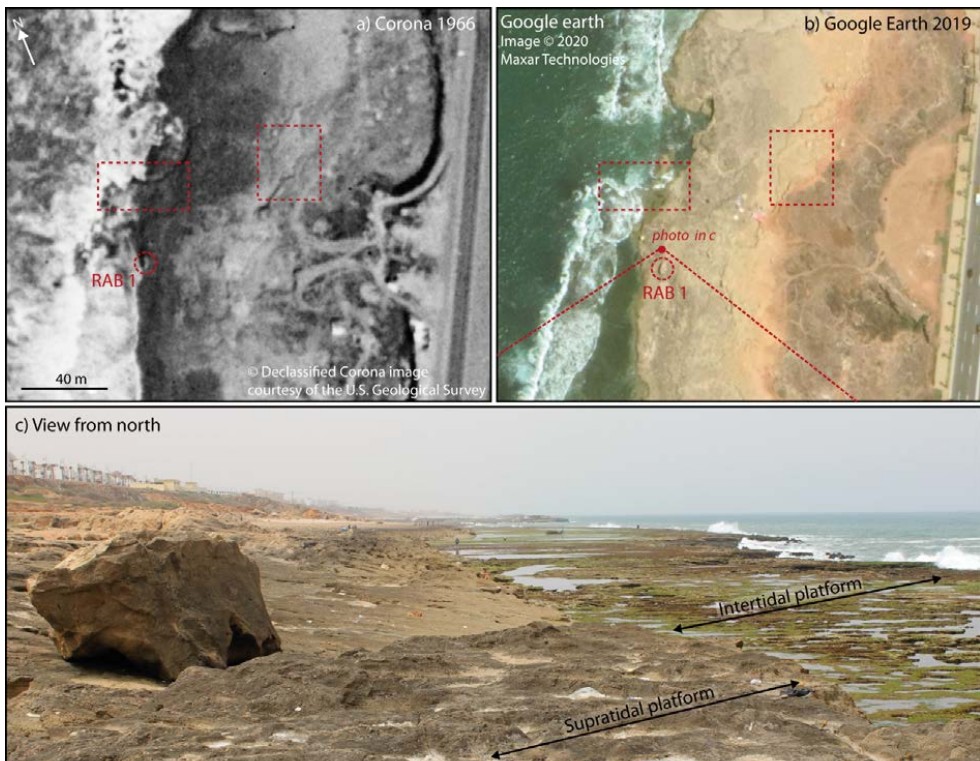

**Fig. A5: Boulder RAB 1. a) RAB 1 (red circle) can be located on the 1966 Corona satellite image. Compared to its present position on the 2019 Google Earth image (b) it might have been pushed a few meters landward but there is no indication of overturning (red rectangles mark features clearly identified on both images for better orientation). c) View towards south with boulder RAB 1 lying on the slope of the supratidal platform (photography July 2016).**

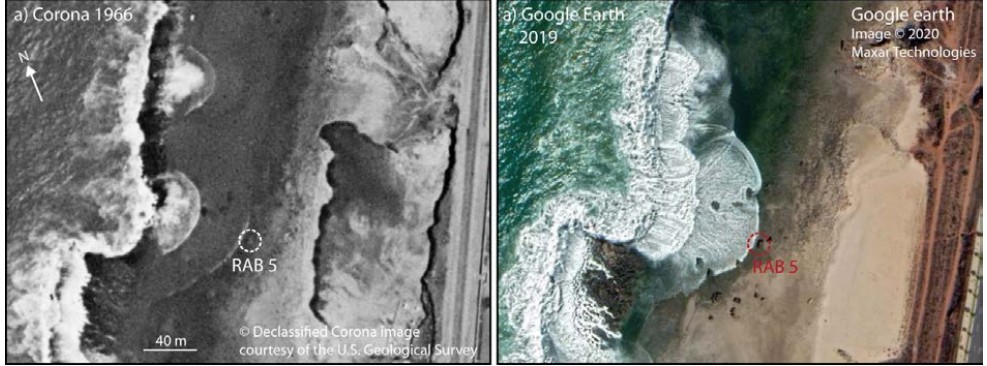

**Fig. A6: Boulder RAB 5. a) RAB 5 (white/red circle) can be located on the 1966 Corona satellite image. It has not changed compared to its present position on the 2019 Google Earth image (b).**





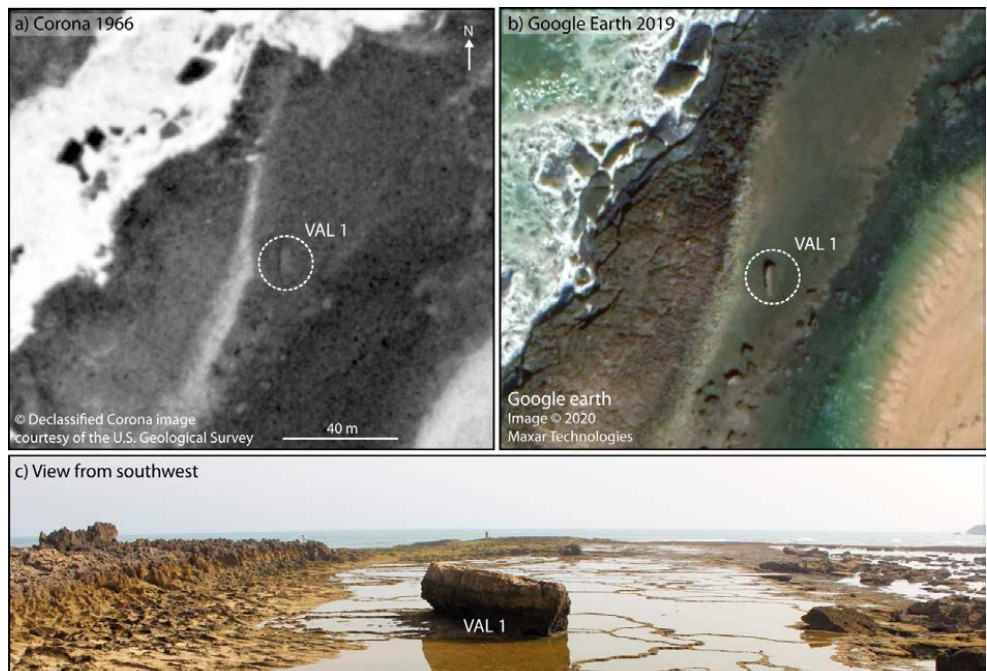

Fig. A7: Boulder VAL 1. a) VAL 1 (white circle) can be located on the 1966 Corona satellite image. It has not changed compared to its present position on the 2019 Google Earth image (b). c) View towards northeast with boulder VAL 1 lying on the intertidal platform behind the youngest ridge (photography July 2016).

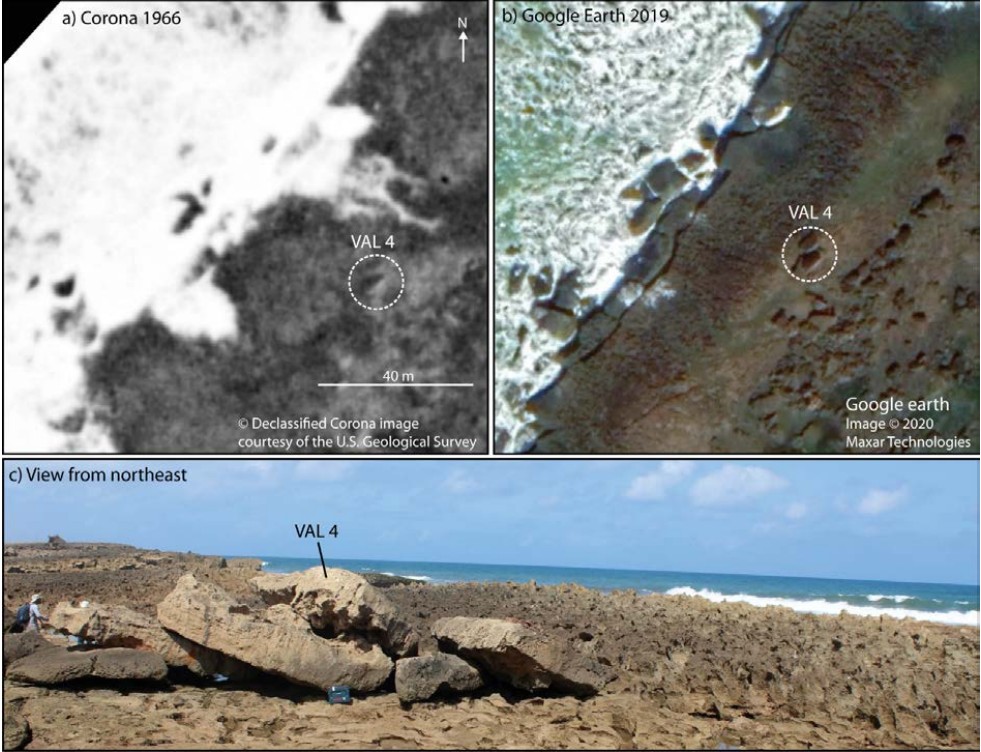





Earth **Surface**
Dynamics
Discussions

**Fig. A8: Boulder VAL 4. a) VAL 4 (white circle) can be located on the 1966 Corona satellite image. It has not changed compared to its present position on the 2019 Google Earth image (b). c) View towards southwest with boulder VAL 4 lying on the youngest ridge (photography July 2016).**

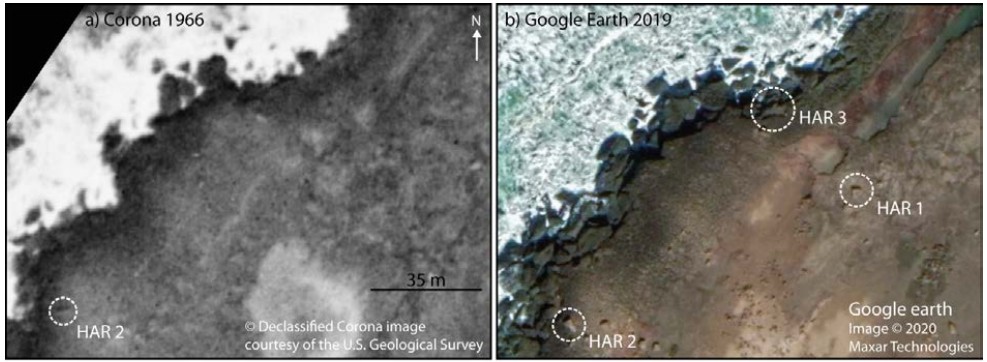

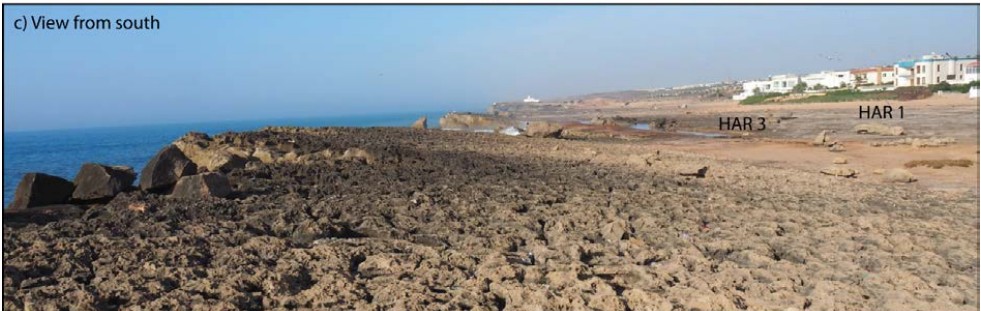

**Fig. A9: Boulders at site HAR. a) Boulder HAR 2 (white circle) can be located on the 1966 Corona satellite image. It slightly rotated along its a-axis, but has not changed its position compared to the 2019 Google Earth image (b). Boulder HAR 1 and niche HAR 3 cannot be identified on the 1966 image; this may be due to poor quality of the image or since they were formed afterwards. c) View towards the north with boulder HAR 1 lying in the depression behind the youngest ridge (photography September 2018).**

Earth **Surface**
Dynamics
Discussions

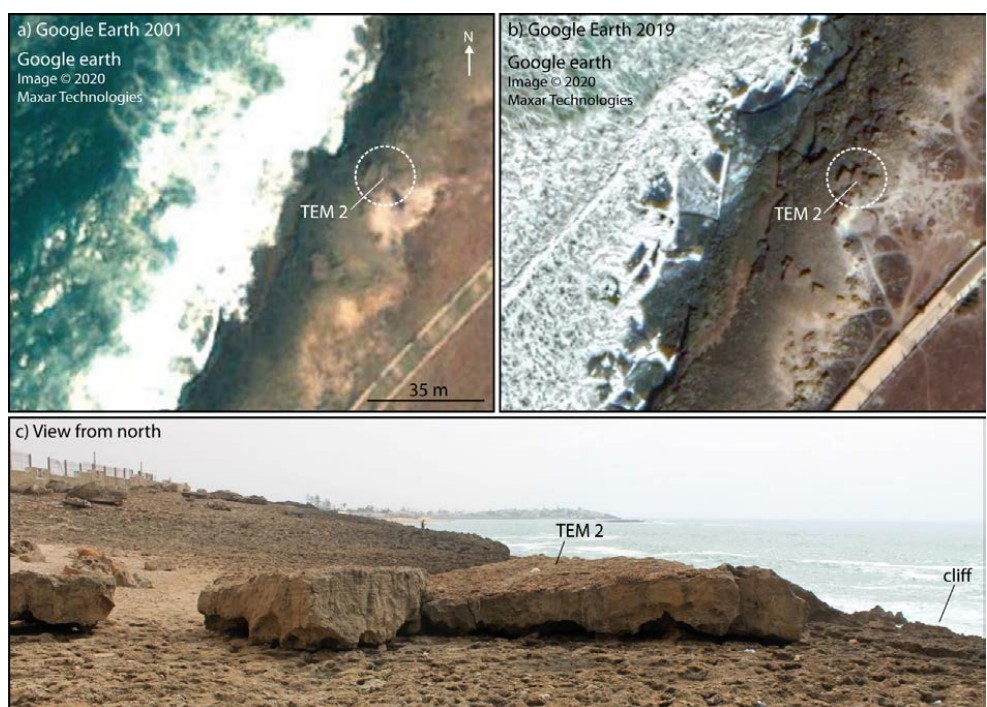

**Fig. A10: Boulder TEM 2 at Temara. a) Boulder TEM 2 (white circle) can be located on 2001 Google Earth satellite images. It has not changed compared to its present position on the 2019 Google Earth image (b). It cannot be identified on the 1966 image; this may be due to poor quality of the image, or since it was deposited afterwards. c) View towards south with boulder TEM 2 lying on the supratidal clifftop platform formed by the youngest ridge (photography Juli 2016).**

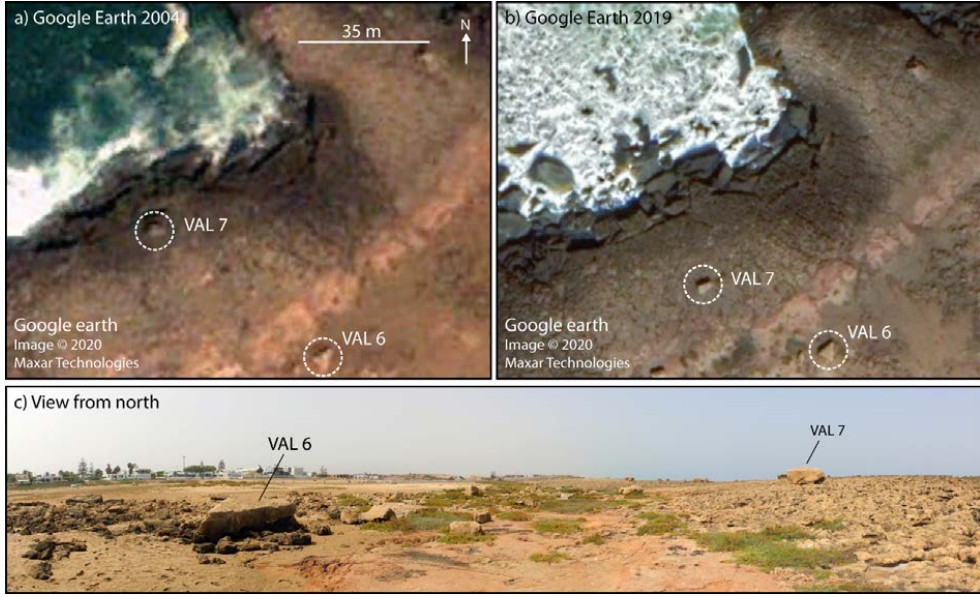

**Fig. A11: Boulder VAL 6. a) Boulder VAL 6 (white circle) can be located on 2004 Google Earth satellite images. It has not changed compared to its present position on the 2019 Google Earth image (b). It cannot be identified on the 1966 Corona image; this may be due to poor quality of the image, or since it was deposited afterwards. c) View towards south with boulder VAL 6 lying in the depression behind the youngest ridge (photography Juli 2016).**



Earth **Surface**
**Dynamics**
Discussions

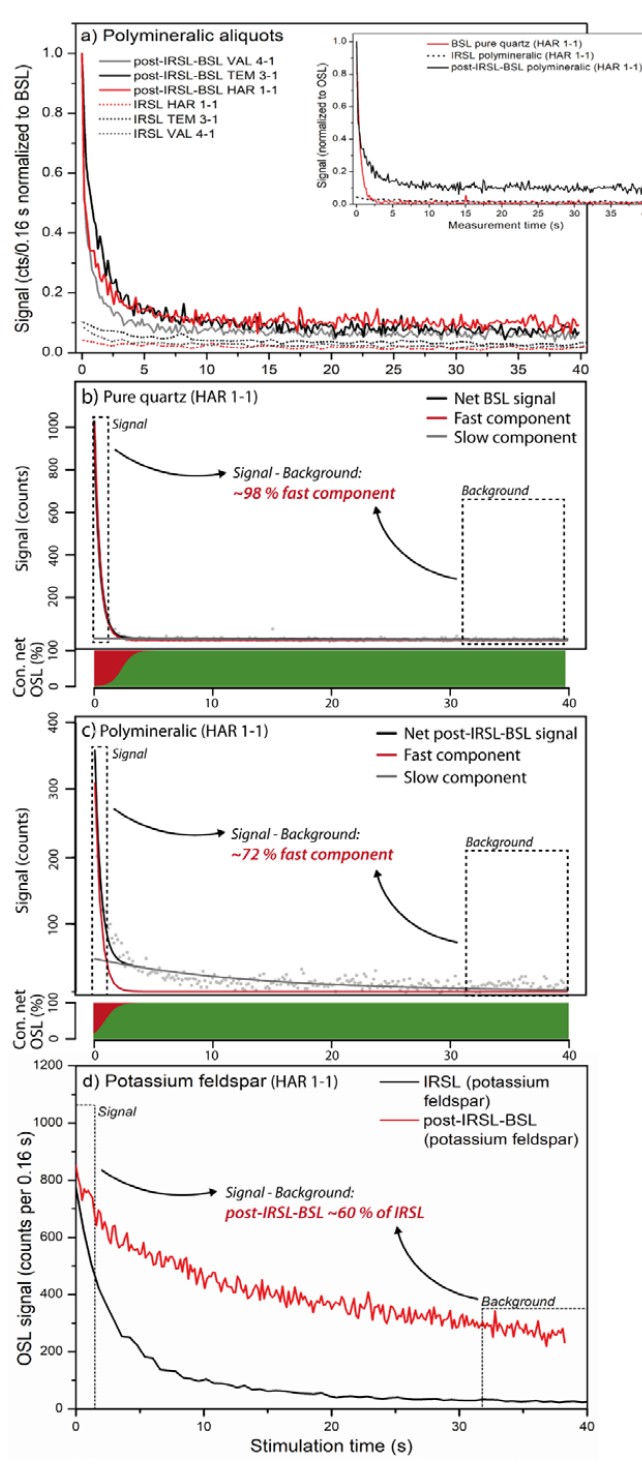

**Fig. A12: Luminescence signal properties in the dated sandstone. a) Decay curves of post-IRSL-BSL and IRSL signals of polymineralic aliquots. The post-IRSL-BSL signals are significantly more intensive than the associated IRSL signals**





**(intensities sum up to only 12-24 % of those of the associated post-IRSL-BSL signals). Inset: Comparison of post-IRSL-BSL and IRSL decay curves of polymineralic aliquots and a pure quartz extract of sample HAR 1-1. b) Quartz BSL signal components achieved by fitting the BSL decay curve of pure quartz of HAR 1-1. The signal in the selected integration limits is dominated by a stable and easily bleachable fast component ($\sigma = 2.4$-$2.5 \times 10^{-17}$, cf. Jain et al., 2003), accounting for 98 % of the analysed net signal. c) Although less pronounced, post-IRSL-BSL signals of polymineralic samples are still dominated by the fast component ($\sigma = 2.4$-$2.5 \times 10^{-17}$, 72% of net signal). d) Comparison of IRSL and post-IRSL-BSL signals measured on potassium feldspar extracts of HAR 1-1. The counts of the background-corrected post-IRSL-BSL signal equal ~60 % of the background-corrected IRSL signal. This indicates that post-IRSL-BSL signals on our polymineralic aliquots are relatively unaffected by a feldspar signal contribution: IRSL signals amount to 12-24 % of the post-IRSL-BSL signals n polymineralic aliquots; 60 % of this IRSL emission still contributes to the post-IRSL-BSL signals, which equals 7.5-15 % of the net post-IRSL-BSL signal in polymineralic aliquots.**

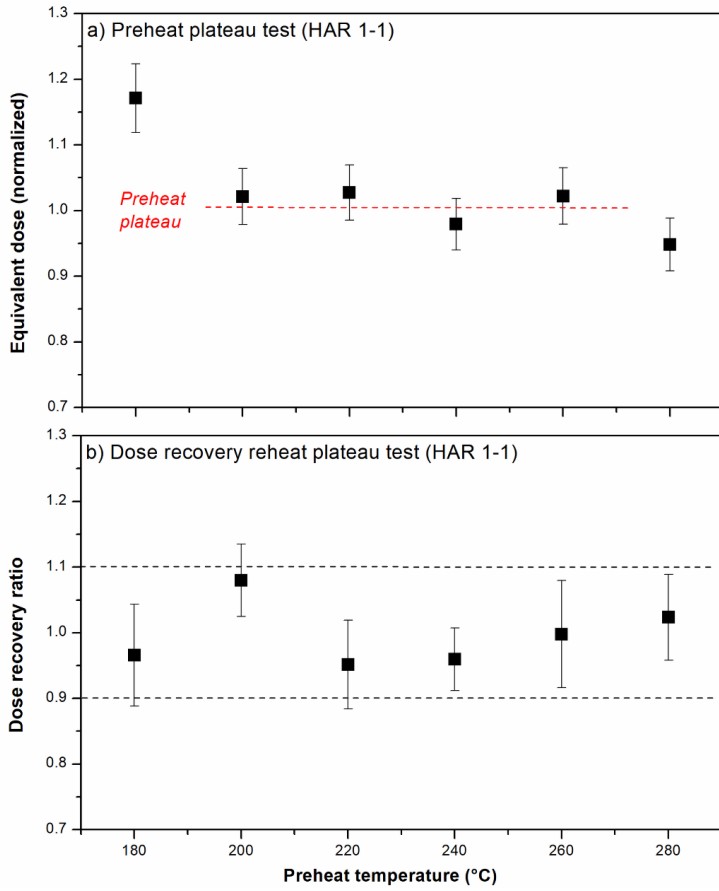

**Fig. A13: Preheat plateau test (a) and dose recovery preheat plateau test with laboratory doses of ~5 Gy (b) performed on quartz extracts of sample HAR 1-1. Both experiments indicate a preheat plateau for temperatures between 200 and 260 °C.**





| Step | Treatment | Signal |
|------|-----------|--------|
| 1 | Preheat (220 °C for 10 s) | |
| 2 | IR LEDs (150 s @ 50 °C) | $L_n$ (IRSL) |
| 3 | Blue LEDs (40 s @ 125 °C) | $L_n$ (post-IRSL-BSL) |
| 4 | Test dose (~12 Gy) | |
| 5 | Preheat (220 °C for 10 s) | |
| 6 | IR LEDs (150 s @ 50 °C) | $T_n$ (IRSL) |
| 7 | Blue LEDs (40 s @ 125 °C) | $T_n$ (post-IRSL-BSL) |

**Tab. A2: Double SAR protocol used for measurement of Ln/Tn data from polymineralic aliquots of crushed slices.**

Earth **Surface**
**Dynamics**
Discussions

EGU



**Fig. A14: Post-IRSL-BSL signal-depth curves for all cores of boulder samples RAB 1-1, RAB 5-1, HAR 1-1, HAR 2-1, HAR 3-1 and TEM 2-1. Black squares = cores included in calculation of mean signal-depth curves, grey circles = cores excluded from calculation of mean signal-depth curves, red squares = mean values.**





**Fig. A15: Post-IRSL-BSL signal-depth curves for all cores of boulder samples TEM 3-1, TEM 4-1, VAL 1-1, VAL 1-2, VAL 4-1 and VAL 6-1. Black squares = cores included in calculation of mean signal-depth curves, grey circles = cores excluded from calculation of mean signal-depth curves, red squares = mean values.**

Earth **Surface**
**Dynamics**
Discussions



**Fig. A16: IRSL signal-depth curves for all cores of boulder samples with adequate feldspar signals (RAB and TEM).**
**Black squares = cores included in calculation of mean signal-depth curves, grey circles = cores excluded from**
**calculation of mean signal-depth curves, red squares = mean values.**





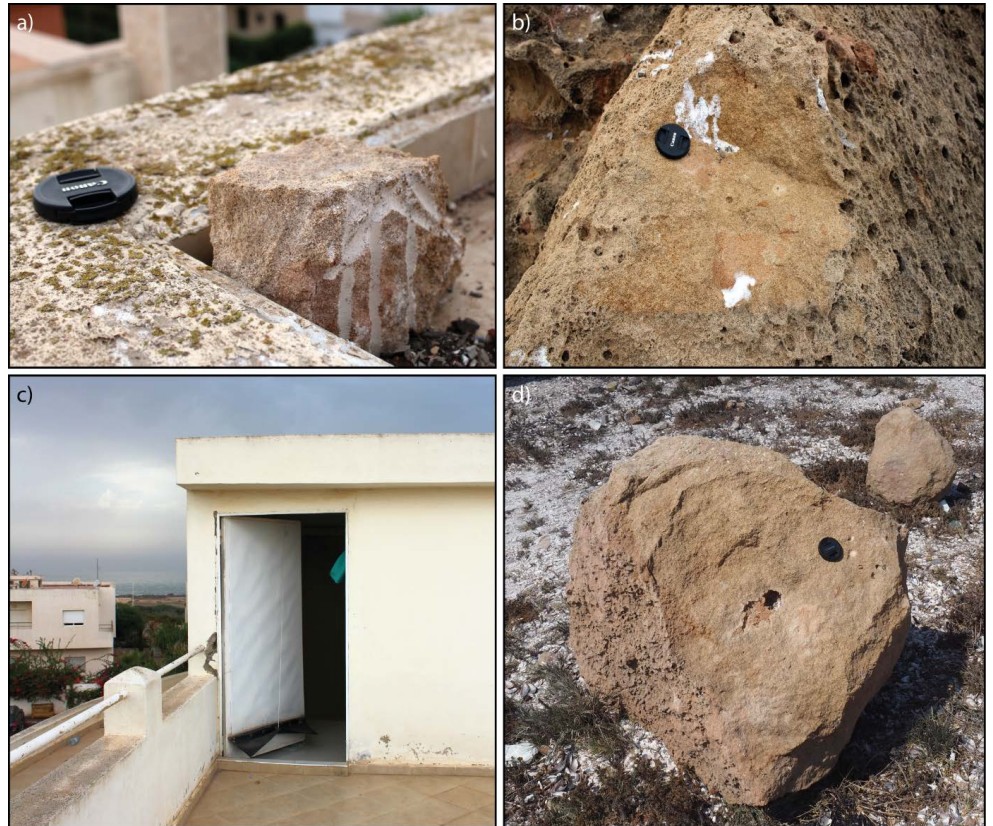

**Fig. A17: Calibration samples. a) Roof top sample VAL 4-1 CAL I. b) Surface on boulder VAL 4 exposed during first field survey in July 2016 by removing at least 10 cm of rock. c) Roof of the house used for artificially exposing rock samples; samples VAL 4-1 CAL I and HAR 1-1 CAL were placed on top of the highest roof shown in the photo. d) Surface of boulder TEM 3 exposed during the first field survey in July 2016; at least 10 cm of rock were removed.**

Earth **Surface**
**Dynamics**
Discussions

EGU



Fig. A18: Post-IRSL-BSL signal-depth curves for all cores of the calibration samples. Black squares = cores included in calculation of mean signal-depth curves, grey circles = cores excluded from calculation of mean signal-depth curves, red squares = mean values.



Earth **Surface**
**Dynamics**
Discussions

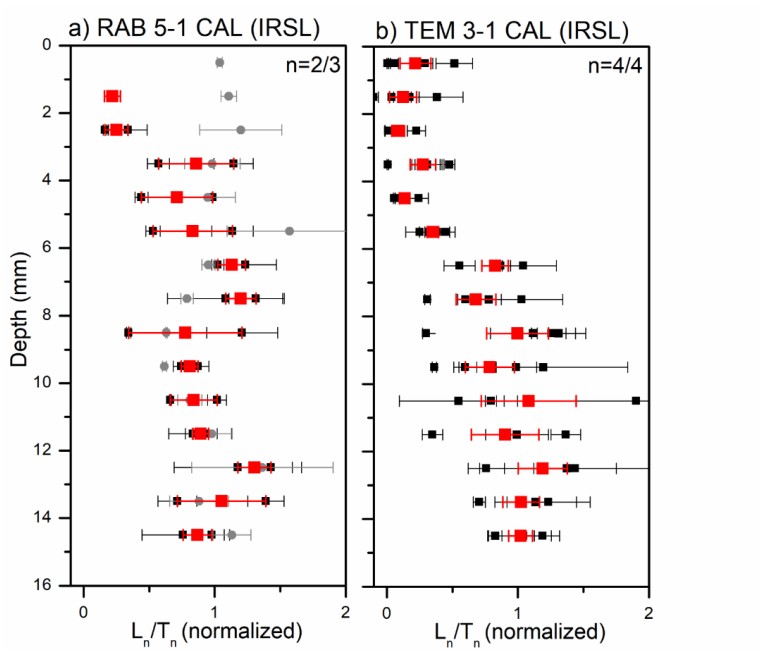

**Fig. A19: IRSL signal-depth curves for all cores of calibration samples with adequate feldspar signals (RAB 5-1 CAL and TEM 3-1 CAL). Black squares = cores included in calculation of mean signal-depth curves, grey circles = cores excluded from calculation of mean signal-depth curves, red squares = mean values.**

| Step | Treatment | Signal |
|---|---|---|
| 1 | Preheat (220 °C for 10 s) | |
| 2 | Blue LEDs (40 s @ 125 °C) | $L_x$ (BSL) |
| 3 | Test dose (~6 Gy) | |
| 4 | Preheat (220 °C for 10 s) | |
| 5 | Blue LEDs (40 s @ 125 °C) | $T_x$ (BSL) |
| 6 | Regenerative dose (R1 to R4, R0, R1) | |
| 7 | Return to Step 1 | |

**Tab. A3: SAR protocol used for equivalent dose measurement of quartz extracts.**



Earth **Surface**
**Dynamics**
Discussions



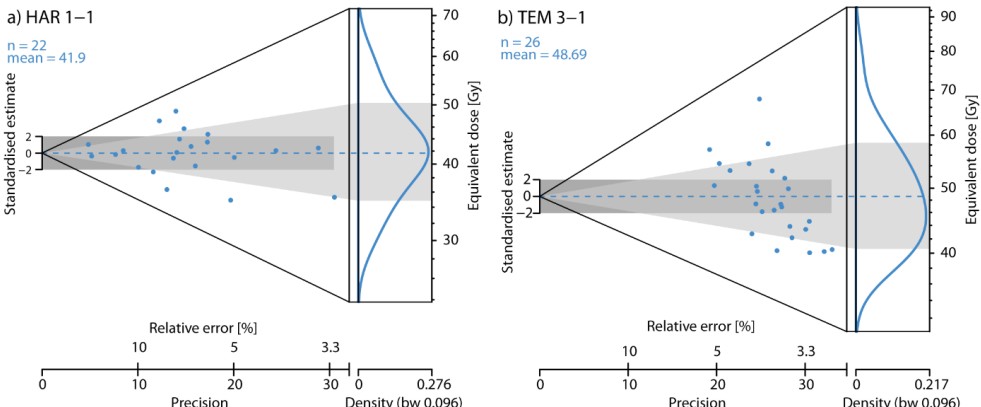

**Fig. A20: Equivalent dose distributions determined on quartz extracts of samples HAR 1-1 and TEM 3-1 (presented as Abanico plots).**

| Sample | U (ppm) | Th (ppm) | K (%) | Dose rate (Gy/ka) | Grain size (µm) | N | OD (%) | CAM De (Gy) | Age (ka) |
|---|---|---|---|---|---|---|---|---|---|
| HAR 1-1 | 0.91±0.06 | 0.62±0.06 | 0.11±0.01 | 0.53±0.02 | 100-200 | 22 | 16±3 | 41.7±1.6 | 81.0±4.1 |
| TEM 3-1 | 0.86±0.05 | 0.62±0.05 | 0.10±0.01 | 0.51±0.02 | 100-200 | 26 | 17±2 | 48.6±1.7 | 98.1±4.8 |

**Tab. A4: Dose rates, equivalent doses and conventional burial ages for samples HAR 1-1 and TEM 3-1.**

| Sample | Control age (years) | Erosion rate (mm/year) |
|---|---|---|
| VAL 1-1 | 50 | <0.01 |
| VAL 1-1 | 6000 | 0.32 |
| VAL 1-2 | 50 | 0.05 |
| VAL 1-2 | 450 | 0.20 |
| VAL 1-2 | 6000 | 0.40 |
| HAR 1-1 | 15 | 0.06 |
| HAR 2-1 | 50 | 0.06 |
| RAB 1-2 | 50 | 0.13 |

**Tab. A5: Modelling of post-transport erosion for samples with exposure ages that underestimate the minimum control ages (VAL 1-1, VAL 1-2, RAB 1-2, HAR 1-1 and HAR 1-2).**