# Peer review of "Evaluating OSL rock surface exposure dating as a novel approach for reconstructing coastal boulder movement on decadal to centennial timescales"

_Earth Surface Dynamics, 2020_

## Referee Comment (RC1) · Pedro Costa (Referee) · 11 Aug 2020

Dear authors, many thanks for the opportunity to review your manuscript on OSL rock surface exposure dating as a novel approach for reconstructing transport histories of coastal boulders over decennial to centennial timescales.

Your manuscript is very well prepared. It is nicely written and fits perfectly within the scope of the journal. The figures serve their purposes very well. In fact they illustrate with high-quality the reasoning forwarded and facilitates the reader's job because they

are very informative. Nevertheless, their number seems a bit excessive and a couple of them couple be merged (e.g.A14-A15-A16).

The text flows well and, with the exception of very few misspelling words, it is impeccable to read. References seen to be updated and formulas used are properly formatted.

Regarding science, this manuscript focus on one key issue on storm and marine deposits, namely in boulder deposits. It is a known problem to accurately date the transport of these boulders in coastal settings and it is a theme that have constrained the accurate establishment of return periods and hazard assessments in many locations worldwide. The authors used a well-controlled setting within a short-time window of observation which allowed comparison with aerial/satellite imagery. Thus, narrowing time-interval of transport being studied. The concept and the example selected is interesting and very sound. However, several question still remain to be answered. I will raise a few below but first would like to stress that I feel this manuscript clearly addresses a relevant topic and, with the results presented, moves science forward.

The "new" OSL methodology presented is robust and should/needs to be further tested in other locations. A shame we do not have this methodology compared with other dates from other previously studied locations. The fact that is from specific locations clearly puts forward its potential but still leaves some doubts regarding its reliability. It would be interesting to have further direct age comparisons.

One aspect that concerns me is the obvious dependence on mineralogy. Limestone coastal areas will still be a challenge and one that needs to be addressed. Nevertheless, this manuscript clearly points very interesting future research directions.

The mineralogy-dependence is an obvious constrain to this methodology. This is also evident when we have weathering or erosion. There are micro-erosion meters and they should have been used. I am aware erosion meters have slow rates and require a larger time-window of observation, nevertheless the modeled erosion rates represent for me a huge degree of uncertainty that might have been avoided with empirical data.

Furthermore, these rates are highly controlled by lithology, mineralogy and texture. So, this section of the manuscript is valuable but would benefit from a larger discussion on its shortcomings. Furthermore, this is a key issue in the new OSL methodology: before dating the surface, one must very accurately establish the erosion since deposition.

Regarding the study case, it has been widely established that in many coasts along the North Atlantic from Iceland (Etienne and Paris, 2010), Ireland (Cox et al., 2019) to Portugal (Oliveira et al., 2020) boulder deposits are essentially associated with storm events. There are occasional cases where tsunami origin has been discussed but many times with caution. In that sense, the authors should be less bold on lines 470-475 in particular when comparing case studies with multiple dating methodologies with others with a single methodology or even with just a single measurement. So, the dominance of short-lived and frequent storms on the creation and shaping of boulder deposits is natural in particular in areas not so prone to tsunami events like the North Atlantic. This raises the issue of poor and difficult recognition of tsunami boulder deposits except when very specific dates are obtained (which is very difficult) or when size of boulders and its heights allows to disregard storm origin...but even then, there is the possibility of being palaeo-storm signatures of past higher sea-levels. So, to conclude the data provided from the study case reinforces the reasoning above and I recommend the authors to stress this aspects by adding a couple of sentences on this.

My comments are broadly generic. I enjoyed the manuscript and it certainly deserves to be published after minor changes. Once more, many thanks for the opportunity to review this manuscript. Regards Pedro JM Costa

---

## Referee Comment (RC2) · Anonymous Referee #2 · 10 Sep 2020

This study attempts to determine the exposure ages of some large wave-transported boulders at the coast of Rabat, Morocco, using OSL rock surface exposure dating (OSL-RSED). The final exposure ages are however deemed as unreliable (i.e. imprecise and inaccurate) because of large data scatter, resulting in significant fitting uncertainties, and underestimated due to the erosion of boulder surfaces. This is altogether not very surprising, given that neither the selected lithology nor the chosen geomorphic settings are suitable for OSL-RSED technique.

OSL-RSED requires sensitive quartz and feldspar minerals, while the target boulders

in this study are calcarenite, a type of limestone that is predominantly composed of carbonate, which does not have the required luminescent properties for OSL dating. OSL-RSED is also based on the sunlight-driven evolution of mm- to cm-scale luminescence-depth profiles beneath rock surfaces, and is thus very susceptible to the effect of erosion, down to sub-mm scales. Such erosion-sensitive profiles cannot be used to derive reliable surface exposure ages from boulders undergoing wave and bio-erosion at rates of $\sim$1 mm.a-1, as is the case in this study.

While I appreciate the amount of effort the authors have put to overcome the challenges arising from this adverse combination of poor luminescence properties and erosion, I am afraid their manuscript, at its present form, is not rigorous enough to be considered for publication in Esurf. I could consider this study as a useful methodological contribution to the rapidly growing literature on OSL-RSED if the OSL methods were sound and the data were treated properly. But in my view, this is unfortunately not the case here. In the following, I give an account of both conceptual and methodological issues, which particularly seem problematic to me and try to explain how they could be dealt with differently, where possible. In my opinion, the manuscript may only be considered for publication after addressing these issues properly in a new submission.

Geomorphology and process/hazard information:

The application of OSL-RSED to coastal boulders as is shown in Fig. 1 is oversimplified, as it does not take the effect of reworking into account. If storm surges have enough energy to detach fresh boulders from bedrock, it is very likely that they can rework (slide and overturn) the previously detached boulders sitting loose on the beach as well. It is thus quite conceivable to imagine that some of the surfaces have undergone multiple burial and exposure events, and not only a single continuous exposure event after detachment, as is conceptualised in Fig. 1. In this environment however, the dose rates are low and the burial events are too short (because storm events have high frequency and occur on decadal timescales) to leave a record in the shape of the OSL-depth profiles. Thus, an observed OSL-depth profile measures the cumulative

exposure time since the detachment event, and has no record of the subsequent storm events that might have reworked the surface.

Consequently, even in the absence of complications due to e.g. erosion and poor luminescence characteristics, such profiles are not particularly useful for deriving process information in similar geomorphic settings. They cannot be used for reconstructing boulder transport histories (as the title suggests), because they do not have a memory of the burial events, and they are not good proxy for storm events either, because they only record the single event that detached them from the cliff and not any of the subsequent storm events. One could argue that subsequent events of similar or higher energy are expected to pluck fresh blocks that could also be dated in a similar manner to give a chronology for the storm events. In that scenario, one would expect to see an overall trend of longer exposure events (the so-called "transport ages" here) and thus deeper OSL profiles as one moves farther from the coast, because the storms should gradually push the older boulders inland with time. But this does not seem to be the case; at least not here. For example, according to the age control, sample VAL 6 at a distance of ∼80 m from the cliff seems to be younger than sample VAL 4, which is located only ∼25 m from the cliff. This presumably implies that boulder detachment is not merely driven by wave power, but is also controlled by other factors such as joint formation and orientation. This inherent geomorphic character can limit the use of OSL-RSED to derive process/hazard information from coastal boulders.

OSL-RSED data presentation:

I find the presentation of profile data in Figs. 4, A14-16 cluttered and obscure. The mean data points with standard errors include all the information one needs to evaluate the reliability of individual data points and the overall progress of the bleaching front in a given surface. These are also the data points that are fitted to derive either the exposure age or erosion rate. So, in my view, the presentation of individual aliquots and cores in the way it is done in Figs. 4, A14-16 does not provide any useful information and impedes a proper assessment of the quality of the data.
The fits to the profile data that are used to derive the parameter values in Table 2 are not shown. Without the fits, one cannot evaluate their goodness and the reliability of the resulting parameter values.

In order to enable a clear evaluation of the data, my suggestion is to only present the mean data points with standard errors and the fits to the mean data.

OSL-RSED calibration

The data from calibration sample RAB 5-1 CAL in Fig. 5 seem to reach a plateau at $\sim$0.8 and not 1. This makes me wonder i) why this sample was normalised differently and ii) how this apparently different normalisation must have affected the calibration values derived from this sample, and hence the mean calibrated parameter values used to derive the exposure ages/erosion rates. I note that the same (mean) data presented in Fig. A18 seem to have been normalised correctly. This needs to be revised, in case the authors choose the keep this sample in a new analysis of calibration data. Please see my comment below.

The data from calibration samples VAL 4-1 CAL 2 and RAB 5-1 CAL seem to be much more scattered than those from the other samples. Given the goodness (badness?) of the fits to such poor-quality data, I do not think that the parameter values derived from these samples can be deemed as reliable. It is also intriguing that although the data from these samples are much more scattered than those from e.g. sample TEM 3-1 CAL, the relative uncertainties on sample-specific \overline{\sigma\phi_0} values derived from these samples are smaller than the uncertainty on the corresponding value obtained for sample TEM 3-1 CAL.

It is argued that the sample-specific $\mu$ values have "huge uncertainties", and therefore site-specific values of $\mu$ have been derived instead as "a reasonable and necessary compromise". This argument is not supported by the presented data, and is not in accordance with our understanding of $\mu$ as a physical parameter. Firstly, the relative standard deviation (RSD) of sample-specific $\mu$ values derived from the calibration samples

in Fig. 5 is ~34%, while the RSD of the corresponding $\overline{\sigma\phi_0}$ values is ~210%. So, if sample-specific $\mu$ values can be dismissed because of large uncertainties and overdispersion, how can sample-specific $\overline{\sigma\phi_0}$ values, which have even greater uncertainties and are more dispersed, be acceptable and taken as a shared parameter between the calibration samples? Secondly, if $\mu$ is dependent on lithology and all samples come from the same calcarenite bedrock, why not sharing $\mu$ between all the samples from all the sites? There is no evidence (or at least not presented here) that bedrock lithology varies from one site to another, so I cannot really see the logic behind sharing $\mu$ between samples from individual sites, but not between all the samples.

The issues mentioned above make me wonder about the robustness of the calibration approach undertaken here and the reliability of the resulting parameter values. To address these issues, I would reanalyse the calibration data by i) excluding the inferior data of samples VAL 4-1 CAL 2 and RAB 5-1 CAL, and ii) sharing $\mu$ between all samples or leaving it as a free sample-specific parameter in fitting.

Erosion rate modelling

The authors have followed a numerical approach (not "analytical" as is mentioned in line 333) to model the OSL erosion rates. But, the OSL erosion rate equation has an exact analytical solution that is already published (see Sohbati et al., 2018). So, there is no need and no scientific justification for making guesses at the solution numerically as is done here. The parameter values derived from the calibration samples can simply be inserted in the erosion rate equation and fitted to the profiles to give erosion rates.

Minor comments:

Line 17: I suggest "wave-driven" instead of "wave-emplaced". The boulders cannot be "emplaced" by waves and "transported" at the same time.

Lines 48-49: "...these approaches are restricted to certain boulder lithologies and time

scales.". So is OSL RSED; it is largely restricted to lithologies that "contain quartz and/or feldspar" and to timescales of "decades, centuries up to a few millennia" as is mentioned later in lines 61-62.

Line 63: Does the statement "...to reconstruct...tsunami frequency patterns..." imply that the tsunami events are expected to follow some sort of temporal/spatial patterns?

Lines 71-72: Consider to change "...erosion of post-transport exposed boulder surfaces..." to "erosion of boulder surfaces exposed after transportation" or something like that.

Line 79: Add "buried" before "sediment".

Line 94: What Fig. 1 is actually showing is a boulder that is detached from a wave-cut platform and overturned by waves. There is no "transportation" involved in the depicted scenario.

Line 147: I cannot see how 2-3 m-high spring tides can reach and exceed the 5-m high first ridge (as is mentioned in line 154) to flood Oulja. Lines 189-196: The preheat temperature should also be mentioned somewhere in these lines as Table A2 is in the Appendix.

Line 191: The stimulation time in Table A2 is 150 s and not 160 s.

Lines 197-208: I suppose the dose recovery and preheat plateau tests described in this paragraph were carried out to guide decision on the most suitable measurement protocol. In that case, this paragraph must precede the previous paragraph in which the actual measurement protocol is explained.

Line 207: The "burial ages" suddenly appear here. So far, only OSL RSED is discussed. It is also mentioned (in lines 104-105) that the buried sides of the boulders are inaccessible and "not tried in this study". So, speaking of burial ages here is confusing to me. In fact, it is first 60 lines further down in the text (line 267) that a careful reader may find out that what here is referred to as burial age, is actually the rock formation

age, calculated by dating quartz extracts from deep layers within the boulders that have never seen light after rock formation. These should not be confused by boulder surface burial ages.

Line 212: I find the use of the term "background level" inappropriate here. Background level in OSL dating is commonly referred to while discussing the stimulation curves. I suggest "plateau" instead.

Lines 214-215: This sounds to be a subjective and qualitative approach towards removing the outliers, while there are various quantitative methods to identify them. One common approach that could also be used here is to remove those data points that are different than the mean by three standard deviations.

Line 229: Not sure what is meant by "comparable preconditions for sunlight exposure". If the scenario is as simple as shown in Fig. 1, then all the boulders must have experienced comparable conditions (i.e. detachment and overturn). But if they are likely to have been reworked (i.e. moved and turned over multiple times) then it is very difficult to imagine how they could have had comparable exposure conditions.

Line 252: How about "target" instead of "dated"?

Lines 253-256: It is difficult for me to judge this inference by the way the data are presented in Fig. A12. The pure quartz BSL, K-rich feldspar IRSL and polymineral post-IRSL-BSL signals must be normalised and shown on the same graph to enable a direct comparison.

Lines 274-275: I assume that calibration was carried out before fitting the actual data? Please present the steps in data analysis in the logical order.

Line 279: Sohbati et al. (2011) is the correct reference.

Lines 293-295: This is an interesting observation that the calibration sample TEM 3-1 CAL that is collected from an inclined surface yields a $\overline{\sigma\phi_0}$ value that is ~3 orders of magnitude larger than the corresponding values estimated for the

horizontal surfaces. If this conclusion still stands after data reanalysis (see my comments above), it would be useful to report the tilt angle of the surface. At the moment, there is no data on the dependence of $\overline{\sigma\phi_0}$ on the incident angle of solar radiation in the literature.

Line 307: What is meant by "inadequate" here?

Line 333: The approach of Lehmann et al. (2019) is numerical not analytical.

Line 356: "observed" instead of "achieved"?

Line 365: It seems unlikely to me that "mineralogy-induced does rate differences" can result in the observed scatter in data from such samples. Hot minerals such as zircon and K-rich feldspars are rare, if not non-existent, in calcarenite. Meyer et al. (2018) have attributed similar scatters in their data to the presence of opaque minerals and iron hydroxides, which strongly impede the penetration of light with depth. In the absence of any independent evidence, this seems more reasonable to me as en explanation here.

Lines 366-368: I am not sure I follow. How can the aliquot-to-aliquot variation in feldspar content can give rise to additional scatter in profile data? Does it mean that test dose is not adequately correcting for this possible variation? Why not? What is the evidence?

Lines 375-378: While the interpretation that age underestimation could have been caused by unreliable $\overline{\sigma\phi_0}$ values and erosion of the boulder surfaces may be right, it would nevertheless be interesting to see what erosion rates one would get by applying the erosion rate model to samples that do not seem to suffer from age underestimation. The erosion rate of such samples must be negligible compared to the erosion rates of the samples showing age underestimation. This should provide a good basis for your interpretation.

Line 377: Does "inadequate" mean "unreliable" here?

Line 388: What is meant by "environmental factors beyond the exposure time"?

Lines 418-419: It may be worth mentioning here that, in retrospect, IRSL signals were likely to work better than the post-IRSL-BSL signals for these samples.

Line 422: What is considered as "insufficiently bright signals"? If the post-IRSL-BSL signals shown in Fig. A12 are typical for these samples, they are all well above background by more than 3.

Line 431-434: 1) The ages obtained from eroding surfaces are "apparent" surface exposure ages. The fact that they underestimate the expected ages, does not mean that they are inaccurate. They may be accurate, but they simply do not reflect the age of the event of interest. 2) There is no scientific basis to support this general statement that the ages from inclined surfaces are inaccurate. Surfaces can be dated regardless of their orientation provided that suitable calibration samples are available.

---

## Author Comment (AC2) · 14 Oct 2020

**Reply to interactive reviewer comment by reviewer 2 (anonymous)**

While we disagree with most of the conceptual concerns raised by reviewer 2, we however appreciate the detailed comments and suggestions on our manuscript. We will reply separately to each of the concerns below.

This study attempts to determine the exposure ages of some large wave-transported boulders at the coast of Rabat, Morocco, using OSL rock surface exposure dating (OSL-RSED). The final exposure ages are however deemed as unreliable (i.e. imprecise and inaccurate) because of large data scatter, resulting in significant fitting uncertainties, and underestimated due to the erosion of boulder surfaces. This is altogether not very surprising, given that neither the selected lithology nor the chosen geomorphic settings are suitable for OSL-RSED technique.

OSL-RSED requires sensitive quartz and feldspar minerals, while the target boulders in this study are calcarenite, a type of limestone that is predominantly composed of carbonate, which does not have the required luminescent properties for OSL dating. OSL-RSED is also based on the sunlight-driven evolution of mm- to cm-scale luminescence-depth profiles beneath rock surfaces, and is thus very susceptible to the effect of erosion, down to sub-mm scales. Such erosion-sensitive profiles cannot be used to derive reliable surface exposure ages from boulders undergoing wave and bio-erosion at rates of ~1 mm a-1, as is the case in this study.

We will address the 5 major points of criticism separately after this general comment, but we feel it is necessary to reply to this specific conceptual comment on site selection already here:

We fully agree that the boulder lithology and the coastal setting used in this study do not provide circumstances that are ideal for OSL-RSED. However, our reasoning for conducting this study was not to apply OSL-RSED to a geomorphological/geological context with ideal preconditions, but to evaluate the potential of the approach for coastal boulder deposits. These deposits indeed potentially represent an important archive for coastal hazard assessment, but they often lack chronological information to be fully exploited. In the absence of alternative dating approaches (which is the case for numerous boulder fields worldwide), any (even relative) chronological information that might be provided by OSL-RSED is useful, because in many locations it is the only chronological information available. In this study we make a first attempt to evaluate the potential of the approach for coastal boulders in general (please note: this is not a dating study), and this includes to accept the challenging conditions and to document how they affect the reliability of the dating approach.

Therefore, we were completely aware of the rather difficult conditions for OSL-RSED of coastal boulders in general when we started the study, and we selected a site that (although not ideal compared to other geomorphological contexts) offered all indispensable prerequisites for the evaluation of OSL-RSED: A lithology containing quartz and feldspar, unambiguous signs of boulder overturning in their taphonomy, and age control at least for some of the boulders. Boulder sites with more appropriate lithologies for OSL-RSED typically lack clear indication of boulder movement and age control, and coastal boulders with better independent chronologies are typically composed of pure limestone that cannot be used for OSL dating. We realize that the reasoning of site selection may not have been explained explicitly enough in the original submission and will add two sentences on this in the introduction of a revised version.

Although not ideal, the properties of these boulders are not as poor as implied by the reviewer comment. Calcarenites are carbonate-dominated and/or carbonate-cemented sandstones (they are predominantly, i.e. > 50 %, composed of carbonate grains). This means that they can contain up to 50 % non-carbonate grains such as quartz and feldspar. At the Rabat coast, the calcarenites generally do contain sensitive quartz and feldspar. This is shown in our study

using pure quartz and feldspar extracts, and it was already documented in other publications prior to this study, e.g. by Barton et al. (2009, Quaternary Science Reviews).

Furthermore, as to the comment on wave- and bio-erosion on boulder surfaces, we have to note that we explicitly did not sample surfaces that were affected by wave- or bio-erosion (except for one case, VAL 1, to investigate the effects of wave- or bio-erosion) under regular/typical non-storm conditions. The samples that are considered for dating are all well above the zone of wave- and bio-erosion. Erosion of their surfaces is driven by atmospheric weathering of the calcarenite, independent of wave- and bio-erosion. Since we selected apparently smooth surfaces with no clear signs of erosion, the quantification of erosion (which in retrospect is larger than expected at least for some of the surfaces) was one aim of this evaluation study.

While I appreciate the amount of effort the authors have put to overcome the challenges arising from this adverse combination of poor luminescence properties and erosion, I am afraid their manuscript, at its present form, is not rigorous enough to be considered for publication in Esurf. I could consider this study as a useful methodological contribution to the rapidly growing literature on OSL-RSED if the OSL methods were sound and the data were treated properly. But in my view, this is unfortunately not the case here. In the following, I give an account of both conceptual and methodological issues, which particularly seem problematic to me and try to explain how they could be dealt with differently, where possible. In my opinion, the manuscript may only be considered for publication after addressing these issues properly in a new submission.

Our study is meant as a methodological contribution, not a dating paper. We think we can address all methodological concerns in a revised version, why we think it is suitable for publication. In the following, we will address the five main points, on which the criticism is based on.

**Geomorphology and process/hazard information:**

The application of OSL-RSED to coastal boulders as is shown in Fig. 1 is oversimplified, as it does not take the effect of reworking into account. If storm surges have enough energy to detach fresh boulders from bedrock, it is very likely that they can rework (slide and overturn) the previously detached boulders sitting loose on the beach as well. It is thus quite conceivable to imagine that some of the surfaces have undergone multiple burial and exposure events, and not only a single continuous exposure event after detachment, as is conceptualised in Fig. 1. In this environment however, the dose rates are low and the burial events are too short (because storm events have high frequency and occur on decadal timescales) to leave a record in the shape of the OSL-depth profiles. Thus, an observed OSL-depth profile measures the cumulative exposure time since the detachment event, and has no record of the subsequent storm events that might have reworked the surface. Consequently, even in the absence of complications due to e.g. erosion and poor luminescence characteristics, such profiles are not particularly useful for deriving process information in similar geomorphic settings. They cannot be used for reconstructing boulder transport histories (as the title suggests), because they do not have a memory of the burial events.

We agree that we can only date the first overturning event of each boulder and not the subsequent movements. So yes, it is right that OSL-RSED of the boulders cannot be used to reconstruct the multiple transportation events that might have moved them to their final position. This is, however, not because of problems to differentiate multiple overturning events. The boulders targeted in this study have most likely been overturned only once. All of the sampled boulders weigh several tons and have a platy shape, corresponding to FI (i.e., flatness index, Nandasena and Tanaka, 2013) values of >1 or mostly even >2. It is documented in boulder literature that such clasts are usually overturned during storms when detached from the cliff (in this situation storm waves can attack the boulders from below, e.g. Noormets et al.

2004), but that it needs waves with much larger velocities and heights to overturn them once they rest scattered on the supratidal platform (e.g. Nandasena, 2020). The predominant transport mode for a non-cubic subaerial boulder (i.e., such as most boulders in this study, with FI >2) is sliding, not rolling (Imamura et al. 2008; Nandasena and Tanaka, 2013; Liu et al., 2015). While we admit that this could be explained more explicitly in the manuscript (we will do so in a revised version), the current state of the art in boulder transport by storms clearly supports the transport model shown in Figure 1 and contradicts any biasing of our OSL-RSED data by multiple overturning events. Movement of the boulders subsequent to cliff detachment can happen and probably has happened to most of the sampled boulders. But due to the boulder's shape, mass and distance from the cliff, sliding is the most plausible transport mode.

We however admit that the present title indeed may be misleading, and we suggest "Evaluating OSL rock surface exposure dating as a novel approach for reconstructing coastal boulder movement on decadal to centennial timescales" as a new title in a revised version.

They are not good proxy for storm events either, because they only record the single event that detached them from the cliff and not any of the subsequent storm events. One could argue that subsequent events of similar or higher energy are expected to pluck fresh blocks that could also be dated in a similar manner to give a chronology for the storm events. In that scenario, one would expect to see an overall trend of longer exposure events (the so-called "transport ages" here) and thus deeper OSL profiles as one moves farther from the coast, because the storms should gradually push the older boulders inland with time. But this does not seem to be the case; at least not here. For example, according to the age control, sample VAL 6 at a distance of _80 m from the cliff seems to be younger than sample VAL 4, which is located only _25 m from the cliff. This presumably implies that boulder detachment is not merely driven by wave power, but is also controlled by other factors such as joint formation and orientation. This inherent geomorphic character can limit the use of OSL-RSED to derive process/hazard information from coastal boulders.

We completely disagree with this opinion, since it contradicts all research on coastal boulder records. The reviewer's argument is clearly opposed by the existing literature on coastal boulders (see e.g. the latest review by Lau and Autret, 2020 and references therein). Coastal boulders have frequently been used as an archive for long-term tsunami and storm hazard assessment (e.g. Terry et al., 2013 and references therein). Regardless of the dating approach used (mainly radiocarbon, U/Th and ESR dating), all of these studies are based on ages for the initial onshore transport of the boulders, i.e. due to detachment from the cliff/reef or due to lifting from subtidal areas to the supratidal platform (e.g. Zhao et al., 2009; Engel and May, 2012; Araoka et al., 2013; Rixhon et al., 2017). While the data presented in these publications do not allow to date each transportation event and consequently not every storm, they show that (i) this limitation is not restricted to OSL-RSED but an inherent problem of all established dating approaches applicable to coastal boulders; (ii) ages of initial onshore transport can give a good impression of the recurrence patterns of storms/tsunami if sufficient boulders are dated, particularly since with increasing age specific events cannot be discriminated chronologically anyway (the fact that the scenario described by the reviewer is not reflected by the small number of ages presented in this study does not mean that the principles behind it do not generally apply); and (iii) boulder movement is often not controlled exclusively by wave power, but it is typically the dominant factor. This means that using coastal boulder records for reconstructing the history of extreme wave events may be limited by some of your concerns, but since they are the best (and often only) archive available for the reconstruction of storm/tsunami impact over geological timescales, these limitations (which apply to all dating approaches, not only OSL-RSED) are widely accepted.

To sum up our reply to the general conceptual issues, it is particularly the potential of OSL-RSED that makes it a promising candidate for providing chronological information on non-limestone, quartz- and/or feldspar-bearing boulder deposits and to make use of the coarse clast record for reconstructing extreme event histories. The exposure dating has also the

potential to provide depositional ages, which is preferred in comparison to dating of marine organisms prone to reworking. We consider this a chance to explore the coastal coarse clast record, and this paper shall present a step forward by evaluating and testing the potential. As we have argued before, the conceptual concerns of reviewer 2 are unsubstantiated.

**OSL-RSED data presentation:**

I find the presentation of profile data in Figs. 4, A14-16 cluttered and obscure. The mean data points with standard errors include all the information one needs to evaluate the reliability of individual data points and the overall progress of the bleaching front in a given surface. These are also the data points that are fitted to derive either the exposure age or erosion rate. So, in my view, the presentation of individual aliquots and cores in the way it is done in Figs. 4, A14-16 does not provide any useful information and impedes a proper assessment of the quality of the data.

The fits to the profile data that are used to derive the parameter values in Table 2 are not shown. Without the fits, one cannot evaluate their goodness and the reliability of the resulting parameter values.

In order to enable a clear evaluation of the data, my suggestion is to only present the mean data points with standard errors and the fits to the mean data.

Thank you for this comment, we understand the criticism of the way the OSL signal-depth data of the individual samples is presented. Our reasoning for presenting the data the way it is done in the original submission was to show the reader the entire data set he analyses is based on. We, however, realize that this may rather distract from the important information, which are the mean values and the fit of the data. In a revised version we will therefore follow the suggestion of reviewer 2 to adjust Figures A14-16 by (i) presenting only average values for each depth, and (ii) plotting the associated fit of the data to allow evaluation of its reliability.

**OSL-RSED calibration:**

The data from calibration sample RAB 5-1 CAL in Fig. 5 seem to reach a plateau at _0.8 and not 1. This makes me wonder i) why this sample was normalised differently and ii) how this apparently different normalisation must have affected the calibration values derived from this sample, and hence the mean calibrated parameter values used to derive the exposure ages/erosion rates. I note that the same (mean) data presented in Fig. A18 seem to have been normalised correctly. This needs to be revised, in case the authors choose the keep this sample in a new analysis of calibration data. Please see my comment below.

Sorry for the confusion. There seems to be a mistake in the axis configuration of this sample in Figure 5, which will be corrected in a revised version. The data set used for the calibration was based on values normalized to 1.0 as it is shown in Figure A18, so the calibration results are not affected by this issue.

The data from calibration samples VAL 4-1 CAL 2 and RAB 5-1 CAL seem to be much more scattered than those from the other samples. Given the goodness (badness?) of the fits to such poor-quality data, I do not think that the parameter values derived from these samples can be deemed as reliable. It is also intriguing that although the data from these samples are much more scattered than those from e.g. sample TEM 3-1 CAL, the relative uncertainties on sample-specific sigmaphi_0 values derived from these samples are smaller than the uncertainty on the corresponding value obtained for sample TEM 3-1 CAL.

It is absolutely right that these two samples are much more scattered than the others and we agree that individual values fitted using the data are not reliable. We therefore only used them

in combination with the two other samples with flat surfaces to fit mutual sigmaphi_0 values. It is nevertheless a good idea to test a calibration of sigmaphi_0 without these samples (as suggested below).

It is argued that the sample-specific μ values have "huge uncertainties", and therefore site-specific values of μ have been derived instead as "a reasonable and necessary compromise". This argument is not supported by the presented data, and is not in accordance with our understanding of μ as a physical parameter.

Firstly, the relative standard deviation (RSD) of sample-specific μ values derived from the calibration samples in Fig. 5 is ~34%, while the RSD of the corresponding sigmanphi_0 values is ~210%. So, if sample-specific μ values can be dismissed because of large uncertainties and overdispersion, how can sample-specific sigmaphi_0 values, which have even greater uncertainties and are more dispersed, be acceptable and taken as a shared parameter between the calibration samples?

Secondly, if μ is dependent on lithology and all samples come from the same calcarenite bedrock, why not sharing μ between all the samples from all the sites? There is no evidence (or at least not presented here) that bedrock lithology varies from one site to another, so I cannot really see the logic behind sharing μ between samples from individual sites, but not between all the samples.

We cannot really follow the argument in this comment. We do not use sample-specific sigmaphi_0 values for calibration. We use a mutual value for all samples (otherwise we would need individual calibration samples for each targeted boulder). Thus we follow the same approach as for μ, i.e. improving the reliability of fitting by sharing the same value for several samples.

While mutual sigmaphi values are, according to current knowledge, a realistic assumption for boulder surfaces from the same area and with the same surface inclination, mutual μ values indeed do not reflect the heterogeneity of rocks even from the same lithological formation (e.g. Gliganic et al., 2019). This is also the case for the study site. Although the lithology is generally similar (all calcarenite) for all boulders targeted in this study, it is not completely uniform along the entire coastline. There are slight differences in granulometry and content of bioclasts. As we explain in the original manuscript version, the best way to account for expected differences in lithology would be to use a sample-specific μ value for each sample. This is, however, impeded by fitting uncertainties, which lead to unreliable sample-specific values. We therefore have to use several samples to derive a mutual μ value. While the lithology is certainly also slightly different between the boulders at each site investigated here, these differences are considered negligible. The more significant differences exist between the different study sites. To account for these rather significant lithological differences between the study sites (for each of them a sufficient number of samples is available), we decided for site-specific μ values. We realized that this is not explicitly mentioned in the original submission and will add some information in a revised version. We, however, also checked the use of a mutual μ value for all samples.

The issues mentioned above make me wonder about the robustness of the calibration approach undertaken here and the reliability of the resulting parameter values. To address these issues, I would reanalyse the calibration data by i) excluding the inferior data of samples VAL 4-1 CAL 2 and RAB 5-1 CAL, and ii) sharing μ between all samples or leaving it as a free sample-specific parameter in fitting.

According to our replies above, we reanalysed the calibration data. (1) We excluded the strongly scattered samples VAL 4-1 CAL 2 and RAB 5-1 CAL when calibrating μ and sigaphi_0. (2) In addition to the site-specific μ values used in the original version of the manuscript, we also checked the use of a mutual μ value of 1.39±0.15 mm$^{-1}$ for all boulders. While these

modifications change the individual ages of each boulder, the overall chronological pattern of the boulders and, thus, our main conclusions are not affected.

**Erosion rate modelling:**

The authors have followed a numerical approach (not "analytical" as is mentioned in line 333) to model the OSL erosion rates. But, the OSL erosion rate equation has an exact analytical solution that is already published (see Sohbati et al., 2018). So, there is no need and no scientific justification for making guesses at the solution numerically as is done here. The parameter values derived from the calibration samples can simply be inserted in the erosion rate equation and fitted to the profiles to give erosion rates.

The approach of Lehmann et al. (2019) that we applied to our samples is indeed a numerical approach. We will correct the wording in the revised version of the manuscript.

We, however, completely disagree that the application of a numerical approach lacks scientific justification while an analytical approach exists. We are aware that an analytical solution for the quantification of erosion from OSL rock surface data was already presented by Sohbati et al. (2018). The numerical approach of Lehmann et al. (2019) that is used in this study was later published in Earth Surface Dynamics, acknowledging the analytical approach but providing an alternative solution for the erosion problem. Both approaches have their advantages and there is no approach that is absolutely superior compared to the other. The analytical solution of Sohbati et al. (2018) might be more elegant and faster, but the numerical approach chosen in this study (which is not guessing, but inferring results from our data) is able to resolve the problem in time and provides a quantification of misfits and thus uncertainties on the results.

**Minor comments:**

Line 17: I suggest "wave-driven" instead of "wave-emplaced". The boulders cannot be "emplaced" by waves and "transported" at the same time.

Thank you for this suggestion. The wording will be changed to "wave-driven" in a revised version.

Lines 48-49: "...these approaches are restricted to certain boulder lithologies and time scales.". So is OSL RSED; it is largely restricted to lithologies that "contain quartz and/or feldspar" and to timescales of "decades, centuries up to a few millennia" as is mentioned later in lines 61-62.

Thank you for this comment. We realized that we have to be more specific here. Palaeomagnetic dating still suffers from a number of intrinsic methodological limitations, and cosmogenic nuclide dating typically cannot provide sufficient resolution on Late Holocene time scales and is, therefore, of limited benefit for the vast majority of coastal boulders. We will add a sentence explaining these details in a revised version of the manuscript.

Line 63: Does the statement "...to reconstruct...tsunami frequency patterns..." imply that the tsunami events are expected to follow some sort of temporal/spatial patterns?

Yes, tsunamis typically show temporal patterns if they are generated by earthquakes. Since the 1755 Lisbon tsunami was triggered by an offshore earthquake, it is not unlikely that potential predecessors follow a certain temporal pattern that is controlled by the accumulation of seismic strain.

Lines 71-72: Consider to change "...erosion of post-transport exposed boulder surfaces..." to "erosion of boulder surfaces exposed after transportation" or something like that.

Thank you for this suggestion. The wording will be changed accordingly.

Line 79: Add "buried" before "sediment".

Will be changed as suggested.

Line 94: What Fig. 1 is actually showing is a boulder that is detached from a wave-cut platform and overturned by waves. There is no "transportation" involved in the depicted scenario.

The relocation of the boulder from the cliff edge to the supratidal coastal platform in an overturning movement clearly involves transportation. In Figure 1 the process of overturning during transport is illustrated by showing two successive stages of boulder movement. We nevertheless propose to change the wording in a revised version to better express the fact that we always date the cliff detachment of overturned boulders and not potential transport events following afterwards (which typically take place as a sliding movement for plate-shaped boulders as selected in this study).

Line 147: I cannot see how 2-3 m-high spring tides can reach and exceed the 5-m high first ridge (as is mentioned in line 154) to flood Oulja.

While the first calcarenite ridge shows average heights of about 5 m above sea level, this barrier occasionally shows sections with lower elevations or can even be breached at river mouths. This is where water can enter the depression of the Oulja during high tides.

Lines 189-196: The preheat temperature should also be mentioned somewhere in these lines as Table A2 is in the Appendix.

Thank you for the suggestion. In a revised version, we will mention the preheat conditions (220 °C, 10 s) in this section.

Line 191: The stimulation time in Table A2 is 150 s and not 160 s.

Sorry for this mistake. This should be 160 s as stated in the main text.

Lines 197-208: I suppose the dose recovery and preheat plateau tests described in this paragraph were carried out to guide decision on the most suitable measurement protocol. In that case, this paragraph must precede the previous paragraph in which the actual measurement protocol is explained.

Thank you for the suggestion. We changed the order of arguments to clarify that these experiments were used as a basis for final protocol selection.

Line 207: The "burial ages" suddenly appear here. So far, only OSL RSED is discussed. It is also mentioned (in lines 104-105) that the buried sides of the boulders are inaccessible and "not tried in this study". So, speaking of burial ages here is confusing to me. In fact, it is first 60 lines further down in the text (line 267) that a careful reader may find out that what here is referred to as burial age, is actually the rock formation age, calculated by dating quartz extracts from deep layers within the boulders that have never seen light after rock formation. These should not be confused by boulder surface burial ages.

Sorry for the confusion. While the ages indeed reflect the timing of sand grain burial during ridge formation, we agree that the term "burial age" may be ambiguous in this study. To differentiate rock surface burial ages (which were not determined in this study) from conventional OSL dating of the sandstone formation (which we refer to here), we will replace "burial ages" by "ages for sandstone formation" in a revised version.

Line 212: I find the use of the term "background level" inappropriate here. Background level in OSL dating is commonly referred to while discussing the stimulation curves. I suggest "plateau" instead.

We absolutely agree that the term "background" may be misleading in this context. The wording will be changed as suggested.

Lines 214-215: This sounds to be a subjective and qualitative approach towards removing the outliers, while there are various quantitative methods to identify them. One common approach that could also be used here is to remove those data points that are different than the mean by three standard deviations.

Thank you for the suggestion. Our approach was indeed somehow subjective. We therefore revised our rejection criteria as follows: (i) Entire cores were excluded, if they did not show any signs of bleaching with depth, while all other cores from the same sample did. (ii) All other data points were classified as outliers according to a deviation from the mean of more than 2 SD. The data did not change significantly. However, we will use this new data set for all analysis in a revised version.

Line 229: Not sure what is meant by "comparable preconditions for sunlight exposure". If the scenario is as simple as shown in Fig. 1, then all the boulders must have experienced comparable conditions (i.e. detachment and overturn). But if they are likely to have been reworked (i.e. moved and turned over multiple times) then it is very difficult to imagine how they could have had comparable exposure conditions.

We explain the meaning of this term in the second part of this sentence. While all boulders used for this study have been overturned only once (see reply to main comment: the platy boulders used in this study were overturned when detached from the cliff, but moved by sliding only or not at all afterwards), sunlight exposure may also be different due to differential shielding after deposition or due to different exposure angles. We will nevertheless change "preconditions" to "conditions", since this seems more appropriate.

Line 252: How about "target" instead of "dated"?

We appreciate this suggestion and will change as suggested in a revised version.

Lines 253-256: It is difficult for me to judge this inference by the way the data are presented in Fig. A12. The pure quartz BSL, K-rich feldspar IRSL and polymineral post-IRSL-BSL signals must be normalised and shown on the same graph to enable a direct comparison.

We do not agree with this opinion. What we want to document is: (1) Post-IRSL-BSL signals of polymineralic aliquots are significantly stronger than the IRSL signals measured on the same polymineralic aliquots. This is documented in Fig. A12a, where normalized values of both signals are compared in the same plot (as asked for by the reviewer). (2) The post-IRSL-BSL signals of polymineralic aliquots are dominated by a quartz signal with only minor influence of feldspar signals. This is documented in Fig. A12d, which shows that the IRSL stimulation used in our protocol reduces the potassium feldspar signal to 60% of its initial value (more details are given in the caption of Fig. A12).

Lines 274-275: I assume that calibration was carried out before fitting the actual data? Please present the steps in data analysis in the logical order.

We use this sentence as an introduction to the explanation, why calibration is necessary. To avoid confusion, we will change the wording to "To estimate boulder ages with OSL-RSED, measured post-IRSL-BSL signal-depth data must be fitted with the bleaching model described in Equation (1)".

Line 279: Sohbati et al. (2011) is the correct reference.

Thank you for the correction. The reference will be changed accordingly.

Lines 293-295: This is an interesting observation that the calibration sample TEM 3-1 CAL that is collected from an inclined surface yields a sigmanphi_0 value that is ~3 orders of magnitude larger than the corresponding values estimated for the horizontal surfaces. If this conclusion still stands after data reanalysis (see my comments above), it would be useful to report the tilt angle of the surface. At the moment, there is no data on the dependence of sigmanphi_0 on the incident angle of solar radiation in the literature.

After reanalysing the data by excluding the two calibration samples with poorly defined bleaching fronts, there is still a significant difference of one order of magnitude between the horizontal calibration samples and the inclined calibration sample (the angle of the surface is already reported in Table 1 with ~25°). We agree that such an observation has not been reported and would be worth a more detailed investigation. We are, however, aware that our assumption is only based on a single sample (and a total set of 5 calibration samples even without excluding RAB 5-1 CAL and VAL 4-1 CAL2). It obviously needs a larger dataset and more controlled conditions (e.g. in a bleaching experiment) to evaluate the assumed relationship between inclination of the surface and sigmaphi_0.

Line 307: What is meant by "inadequate" here?

We will change "inadequate" to "incorrect".

Line 333: The approach of Lehmann et al. (2019) is numerical not analytical.

This is of course correct. Will be changed to "numerical" in a revised version.

Line 356: "observed" instead of "achieved"?

Thank you for the suggestion. The wording will be changed accordingly.

Line 365: It seems unlikely to me that "mineralogy-induced dose rate differences" can result in the observed scatter in data from such samples. Hot minerals such as zircon and K-rich feldspars are rare, if not non-existent, in calcarenite. Meyer et al. (2018) have attributed similar scatters in their data to the presence of opaque minerals and iron hydroxides, which strongly impede the penetration of light with depth. In the absence of any independent evidence, this seems more reasonable to me as an explanation here.

We agree that this argument will definitely not explain most of the observed scatter. While it may add to the observed scatter of signals (that is why we included the argument originally), it is likely of very minor importance and might involuntarily make the discussion more complicated than necessary. We therefore decided to abstain from using this argument in a revised version of the manuscript.

Lines 366-368: I am not sure I follow. How can the aliquot-to-aliquot variation in feldspar content can give rise to additional scatter in profile data? Does it mean that test dose is not adequately correcting for this possible variation? Why not? What is the evidence?

We do not have direct evidence for this argument. We however know that the IRSL stimulation of our post-IRSL-BSL protocol is removing most of the feldspar signal, but not all of it. Although we assume that the feldspar contribution is insignificant based on our test measurements, it must be expected that the contribution of feldspar signals to the post-IRSL-BSL signal will be slightly different for polymineralic aliquots with different percentages of feldspar. Since this

potential source of scatter will again explain (if at all) only a very minor part of the observed scatter, we again decided to abstain from using this argument in a revised version of the manuscript and focus on the most plausible arguments.

Lines 375-378: While the interpretation that age underestimation could have been caused by unreliable sigmaphi values and erosion of the boulder surfaces may be right, it would nevertheless be interesting to see what erosion rates one would get by applying the erosion rate model to samples that do not seem to suffer from age underestimation. The erosion rate of such samples must be negligible compared to the erosion rates of the samples showing age underestimation. This should provide a good basis for your interpretation.

The point developed by reviewer 2 is fair, and we will tackle this question in a revised version of the manuscript.

Line 377: Does "inadequate" mean "unreliable" here?

Yes. We will replace "inadequate" by "unreliable" in the revised version to make this absolutely clear.

Line 388: What is meant by "environmental factors beyond the exposure time"?

The factors refereed to here, i.e. post-transport erosion and occasional shielding of the post-transport surface by e.g. water, are explained in the following sections of the manuscript. We will change the wording to "factors different than exposure time" to avoid any confusion.

Lines 418-419: It may be worth mentioning here that, in retrospect, IRSL signals were likely to work better than the post-IRSL-BSL signals for these samples.

We will add a short reference regarding the potential benefits of IRSL signals for some of our samples: "While IRSL signals were not used in this study due to insufficiently bright signals for most samples, in retrospect their use might be advantageous to post-IRSL-BSL signals at least for some of the investigated samples.

Line 422: What is considered as "insufficiently bright signals"? If the post-IRSL-BSL signals shown in Fig. A12 are typical for these samples, they are all well above background by more than 3.

The term "insufficiently bright" refers to the IRSL signals of polymineralic samples. Those shown in Figure A12a are representative for the samples of the different sites and hardly distinguishable from the background (signal < 3 time background for most aliquots).

Line 431-434: 1) The ages obtained from eroding surfaces are "apparent" surface exposure ages. The fact that they underestimate the expected ages, does not mean that they are inaccurate. They may be accurate, but they simply do not reflect the age of the event of interest. 2) There is no scientific basis to support this general statement that the ages from inclined surfaces are inaccurate. Surfaces can be dated regardless of their orientation provided that suitable calibration samples are available.

We agree with and appreciate these arguments. What we want to express is that apparent ages do not agree with age control due to erosion or unreliable calibration samples. We will change the phrasing of this section to better reflect this argumentation in a revised version.

---

## Author Response (AR1)

**Reply to interactive reviewer comment by reviewer 1 (Pedro Costa)**

We appreciate the constructive comments and suggestions on our manuscript. Below, we will reply to each of them separately (please note: line numbers refer to the revised version with track changes).

Comment: Your manuscript is very well prepared. It is nicely written and fits perfectly within the scope of the journal. The figures serve their purposes very well. In fact they illustrate with high-quality the reasoning forwarded and facilitates the reader's job because they are very informative. Nevertheless, their number seems a bit excessive and a couple of them couple be merged (e.g.A14-A15-A16).

Reply: We decided to use a large number of figures to document our findings to the reader as comprehensive as possible. However, we agree that an excessive use of figures may be rather distracting from the main aspects of the paper and have merged and excluded some of the supplement figures.

Changes in revised version: We have merged figures of the appendix where possible in the revised version. Figs A14 to A16 were simplified by only presenting average values and omitting individual measurement data (as also recommended by reviewer 2). This allows the IRSL data presented in Fig. A16 to be merged with the associated plots in Figs A14 and A15. Figures A18 and A19 have been excluded from the manuscript, since they duplicate data from Figure 5.

Comment: The text flows well and, with the exception of very few misspelling words, it is impeccable to read. References seem to be updated and formulas used are properly formatted.

Regarding science, this manuscript focus on one key issue on storm and marine deposits, namely in boulder deposits. It is a known problem to accurately date the transport of these boulders in coastal settings and it is a theme that have constrained the accurate establishment of return periods and hazard assessments in many locations worldwide. The authors used a well-controlled setting within a short-time window of observation which allowed comparison with aerial/satellite imagery. Thus, narrowing time-interval of transport being studied. The concept and the example selected is interesting and very sound. However, several question still remain to be answered. I will raise a few below but first would like to stress that I feel this manuscript clearly addresses a relevant topic and, with the results presented, moves science forward.

The "new" OSL methodology presented is robust and should/needs to be further tested in other locations. A shame we do not have this methodology compared with other dates from other previously studied locations. The fact that is from specific locations clearly puts forward its potential but still leaves some doubts regarding its reliability. It would be interesting to have further direct age comparisons.

Reply: We absolutely agree that independent age control is required to better evaluate the reliability of the dating approach. Unfortunately, most alternative dating techniques that have been used for determining boulder chronologies so far (i.e. mainly radiocarbon and U/Th dating of coral boulders or attached organisms) are associated with pure limestone lithologies, which cannot be used for OSL dating. Cosmogenic nuclide dating that would work on the same rocks, is not sensitive enough to provide useful age control due to low production rates at sea level and the comparatively short time scales of a few centuries or less. There are currently plans to try to establish a lichen chronometry for the study site, an approach that showed large potential for the time scales we are talking about in a recently published study (Oliveira et al., 2020, Progress in Physical Geography). But even if this attempt should be successful, it will take years to work robustly.

Similar constraints apply to most other boulder deposits. So when we selected the site for this study, we chose boulders with potentially adequate properties for OSL-RSED (which excludes pure limestone boulders due to the lack of quartz and feldspar, and magmatic boulders due to problems with clearly identifying overturning), for which at least age control in form of satellite data and observations for the last decades was available. Since this age control is undoubtedly limited, the presented study is of course only a first attempt to better understand the potential and the challenges associated with the dating approach. More case studies are definitely required to further evaluate the reliability of the dating approach, and we think that the selection of future sites will significantly benefit from the conclusions drawn from our data.

Changes in revised version: We realized that the reasoning for site selection may not have been explained explicitly enough in the original manuscript. In the revised version, we added some explaining sentences to the introduction in lines 72-77.

Comment: One aspect that concerns me is the obvious dependence on mineralogy. Limestone coastal areas will still be a challenge and one that needs to be addressed. Nevertheless, this manuscript clearly points very interesting future research directions.

Reply: Indeed OSL-RSED cannot be applied to pure limestone boulders, which unfortunately excludes a large portion of all boulder deposits, particularly in tropical regions. However, the approach promises to provide chronological information for boulder sites with quartz and/or feldspar bearing lithologies, such as sandstones, calcarenites and igneous boulders, which also account for a significant number of boulder sites. In other words, we do not pretend to present a dating solution that is applicable to all boulder deposits, but a technique that might provide chronological information for some of them. It is, however, important to highlight, that OSL-RSED can address boulders which are specifically hard to date with alternative approaches so far. Most existing chronologies for Holocene boulders are restricted to limestone boulders that are composed of or associated with calcareous organisms datable by radiocarbon or U/Th.

Changes in revised version: We now document the lithology-related limitations and chances of OSL-RSED more explicitly in the introduction (lines 66-68) and conclusions (lines 588-590) of the revised version.

Comment: The mineralogy-dependence is an obvious constrain to this methodology. This is also evident when we have weathering or erosion. There are micro-erosion meters and they should have been used. I am aware erosion meters have slow rates and require a larger time-window of observation, nevertheless the modelled erosion rates represent for me a huge degree of uncertainty that might have been avoided with empirical data. Furthermore, these rates are highly controlled by lithology, mineralogy and texture. So, this section of the manuscript is valuable but would benefit from a larger discussion on its shortcomings. Furthermore, this is a key issue in the new OSL methodology: before dating the surface, one must very accurately establish the erosion since deposition.

Reply: We appreciate the suggestions to improve the discussion of erosion as a key factor for reliable OSL-RSED ages. Micro-erosion meters are a very good idea that we unfortunately did not consider when starting the study, but which should be included in systematic future studies on OSL-RSED as a possible means of better evaluating modelled erosion rates inferred from the OSL data. We already discussed the uncertainties introduced by dating unstable (eroding) surfaces and consider the influence of texture and mineralogy on erosion rates, since these are inherent factors controlling the model output of individual samples. We, however, agree that the paper would also benefit from a critical discussion of the approach we used to determine erosion rates and about benefits of potential alternative approaches, such as erosion meters.

Changes in revised version: We extended the discussion of erosion rates in the revised version of the manuscript by implementing a critical view on the limitations of modelling and the potential benefits of alternative approaches (lines 458-466).

Comment: Regarding the study case, it has been widely established that in many coasts along the North Atlantic from Iceland (Etienne and Paris, 2010), Ireland (Cox et al., 2019) to Portugal (Oliveira et al., 2020) boulder deposits are essentially associated with storm events. There are occasional cases where tsunami origin has been discussed but many times with caution. In that sense, the authors should be less bold on lines 470-475 in particular when comparing case studies with multiple dating methodologies with others with a single methodology or even with just a single measurement. So, the dominance of short-lived and frequent storms on the creation and shaping of boulder deposits is natural in particular in areas not so prone to tsunami events like the North Atlantic. This raises the issue of poor and difficult recognition of tsunami boulder deposits except when very specific dates are obtained (which is very difficult) or when size of boulders and its heights allows to disregard storm origin...but even then, there is the possibility of being palaeo-storm signatures of past higher sea-levels. So, to conclude the data provided from the study case reinforces the reasoning above and I recommend the authors to stress this aspects by adding a couple of sentences on this.

Replay: Thank you for this comment. We agree that the aspect of discriminating between storm and tsunami origin might need a bit longer discussion. In essence, our data support the reviewer's opinion that in most regions the majority of coastal boulders are associated with storms and that a tsunami origin at such locations is usually hard to verify with the chronological data available. As such, our data also support that boulders identified along the Atlantic coasts of Morocco and Iberia have to be treated with caution when it comes to discussing their tsunami origin, since the associated chronologies usually do not allow to precisely differentiate specific events. It is, however, right that most of the associated studies already acknowledge storms as an alternative transport mechanism. We apologize, if our formulation has implied something else.

Changes in revised version: We used a more cautious wording with regard to the interpretation of coastal boulders in other studies in the revised version of the manuscript (lines 558-566). This also includes a brief but more detailed discussion of the difficulties related to tsunami boulder recognition (lines 509-513).

**Reply to interactive reviewer comment by reviewer 2 (anonymous)**

While we disagree with most of the conceptual concerns raised by reviewer 2, we however appreciate the detailed comments and suggestions on our manuscript. We will reply separately to each of the concerns below (note: line numbers refer to the revised version with track changes).

Comment: This study attempts to determine the exposure ages of some large wave-transported boulders at the coast of Rabat, Morocco, using OSL rock surface exposure dating (OSL-RSED). The final exposure ages are however deemed as unreliable (i.e. imprecise and inaccurate) because of large data scatter, resulting in significant fitting uncertainties, and underestimated due to the erosion of boulder surfaces. This is altogether not very surprising, given that neither the selected lithology nor the chosen geomorphic settings are suitable for OSL-RSED technique.

OSL-RSED requires sensitive quartz and feldspar minerals, while the target boulders in this study are calcarenite, a type of limestone that is predominantly composed of carbonate, which does not have the required luminescent properties for OSL dating. OSL-RSED is also based on the sunlight-driven evolution of mm- to cm-scale luminescence-depth profiles beneath rock surfaces, and is thus very susceptible to the effect of erosion, down to sub-mm scales. Such erosion-sensitive profiles cannot be used to derive reliable surface exposure ages from boulders undergoing wave and bio-erosion at rates of ~1 mm a-1, as is the case in this study.

Reply: We will address the 5 major points of criticism separately after this general comment, but we feel it is necessary to reply to this specific conceptual comment on site selection already here.

We fully agree that the boulder lithology and the coastal setting used in this study do not provide circumstances that are ideal for OSL-RSED. However, our reasoning for conducting this study was not to apply OSL-RSED to a geomorphological/geological context with ideal preconditions, but to evaluate the potential of the approach for coastal boulder deposits. These deposits indeed potentially represent an important archive for coastal hazard assessment, but they often lack chronological information to be fully exploited. In the absence of alternative dating approaches (which is the case for numerous boulder fields worldwide), any (even relative) chronological information that might be provided by OSL-RSED is useful, because in many locations it is the only chronological information available. In this study we make a first attempt to evaluate the potential of the approach for coastal boulders in general (please note: this is not a dating study), and this includes to accept the challenging conditions and to document how they affect the reliability of the dating approach.

Therefore, we were completely aware of the rather difficult conditions for OSL-RSED of coastal boulders in general when we started the study, and we selected a site that (although not ideal compared to other geomorphological contexts) offered all indispensable prerequisites for the evaluation of OSL-RSED: A lithology containing quartz and feldspar, unambiguous signs of boulder overturning in their taphonomy, and age control at least for some of the boulders. Boulder sites with more appropriate lithologies for OSL-RSED typically lack clear indication of boulder movement and age control, and coastal boulders with better independent chronologies are typically composed of pure limestone that cannot be used for OSL dating.

Although not ideal, the properties of these boulders are not as poor as implied by the reviewer comment. Calcarenites are carbonate-dominated and/or carbonate-cemented sandstones (they are predominantly, i.e. > 50 %, composed of carbonate grains). This means that they can contain up to 50 % non-carbonate grains such as quartz and feldspar. At the Rabat coast, the calcarenites generally do contain sensitive quartz and feldspar. This is shown in our study using pure quartz and feldspar extracts, and it was already documented in other publications prior to this study, e.g. by Barton et al. (2009, Quaternary Science Reviews).

Furthermore, as to the comment on wave- and bio-erosion on boulder surfaces, we have to note that we explicitly did not sample surfaces that were affected by wave- or bio-erosion (except for one case, VAL 1, to investigate the effects of wave- or bio-erosion) under regular/typical non-storm conditions. The samples that are considered for dating are all well above the zone of wave- and bio-erosion. Erosion of their surfaces is driven by atmospheric weathering of the calcarenite, independent of wave- and bio-erosion. Since we selected apparently smooth surfaces with no clear signs of erosion, the quantification of erosion (which in retrospect is larger than expected at least for some of the surfaces) was one aim of this evaluation study.

195

Changes in the revised version: We realized that the reasoning of site selection may not have been explained explicitly enough in the original submission and, therefore, added two sentences with regard to this topic in the introduction of the revised version (lines 72-77).

Comment: While I appreciate the amount of effort the authors have put to overcome the challenges arising from this adverse combination of poor luminescence properties and erosion, I am afraid their manuscript, at its present form, is not rigorous enough to be considered for publication in Esurf. I could consider this study as a useful methodological contribution to the rapidly growing literature on OSL-RSED if the OSL methods were sound and the data were treated properly. But in my view, this is unfortunately not the case here. In the following, I give an account of both conceptual and methodological issues, which particularly seem problematic to me and try to explain how they could be dealt with differently, where possible. In my opinion, the manuscript may only be considered for publication after addressing these issues properly in a new submission.

Reply: Our study is meant as a methodological contribution, not a dating paper. We think we have addressed all methodological concerns in the revised version, why we think it is suitable for publication. In the following, we will address the five main points, on which the criticism is based on.

**Geomorphology and process/hazard information:**

Comment: The application of OSL-RSED to coastal boulders as is shown in Fig. 1 is oversimplified, as it does not take the effect of reworking into account. If storm surges have enough energy to detach fresh boulders from bedrock, it is very likely that they can rework (slide and overturn) the previously detached boulders sitting loose on the beach as well. It is thus quite conceivable to imagine that some of the surfaces have undergone multiple burial and exposure events, and not only a single continuous exposure event after detachment, as is conceptualised in Fig. 1. In this environment however, the dose rates are low and the burial events are too short (because storm events have high frequency and occur on decadal timescales) to leave a record in the shape of the OSL-depth profiles. Thus, an observed OSL-depth profile measures the cumulative exposure time since the detachment event, and has no record of the subsequent storm events that might have reworked the surface. Consequently, even in the absence of complications due to e.g. erosion and poor luminescence characteristics, such profiles are not particularly useful for deriving process information in similar geomorphic settings. They cannot be used for reconstructing boulder transport histories (as the title suggests), because they do not have a memory of the burial events.

Reply: We agree that we can only date the first overturning event of each boulder and not the subsequent movements. So yes, it is right that OSL-RSED of the boulders cannot be used to reconstruct the multiple transportation events that might have moved them to their final position (we admit that the present title indeed may be misleading). This is, however, not because of problems to differentiate multiple overturning events. The boulders targeted in this study have most likely been overturned only once. All of the sampled boulders weigh several tons and have a platy shape, corresponding to FI (i.e., flatness index, Nandasena and Tanaka, 2013) values of >1 or mostly even >2. It is documented in boulder literature that such clasts are usually overturned during storms when detached from the cliff (in this situation storm waves can attack the boulders from below, e.g. Noormets et al. 2004), but that it needs waves with much larger velocities and heights to overturn them once they rest scattered on the supratidal platform (e.g. Nandasena, 2020). The predominant transport mode for a non-cubic subaerial boulder (i.e., such as most boulders in this study, with FI >2) is sliding, not rolling (Imamura et al. 2008;

Nandasena and Tanaka, 2013; Liu et al., 2015). While we admit that this could be explained more explicitly in the manuscript, the current state of the art in boulder transport by storms clearly supports the transport model shown in Figure 1 and contradicts any biasing of our OSL-RSED data by multiple overturning events. Movement of the boulders subsequent to cliff detachment can happen and probably has happened to most of the sampled boulders. But due to the boulder's shape, mass and distance from the cliff, sliding is the most plausible transport mode.

Changes in the revised version: We changed the title of the manuscript to "Evaluating OSL rock surface exposure dating as a novel approach for reconstructing coastal boulder movement on decadal to centennial timescales" in order to better reflect the limitations of the approach with regard to dating sliding motion after cliff detachment. Furthermore, we improved the description of boulder transport at the study site, to clarify that boulders have been overturned only once (lines 59-63 and lines 106-120).

Comment: They are not good proxy for storm events either, because they only record the single event that detached them from the cliff and not any of the subsequent storm events. One could argue that subsequent events of similar or higher energy are expected to pluck fresh blocks that could also be dated in a similar manner to give a chronology for the storm events. In that scenario, one would expect to see an overall trend of longer exposure events (the so-called "transport ages" here) and thus deeper OSL profiles as one moves farther from the coast, because the storms should gradually push the older boulders inland with time. But this does not seem to be the case; at least not here. For example, according to the age control, sample VAL 6 at a distance of _80 m from the cliff seems to be younger than sample VAL 4, which is located only _25 m from the cliff. This presumably implies that boulder detachment is not merely driven by wave power, but is also controlled by other factors such as joint formation and orientation. This inherent geomorphic character can limit the use of OSL-RSED to derive process/hazard information from coastal boulders.

Reply: We completely disagree with this opinion, since it contradicts all research on coastal boulder records. The reviewer's argument is clearly opposed by the existing literature on coastal boulders (see e.g. the latest review by Lau and Autret, 2020 and references therein). Coastal boulders have frequently been used as an archive for long-term tsunami and storm hazard assessment (e.g. Terry et al., 2013 and references therein). Regardless of the dating approach used (mainly radiocarbon, U/Th and ESR dating), all of these studies are based on ages for the initial onshore transport of the boulders, i.e. due to detachment from the cliff/reef or due to lifting from subtidal areas to the supratidal platform (e.g. Zhao et al., 2009; Engel and May, 2012; Araoka et al., 2013; Rixhon et al., 2017). While the data presented in these publications do not allow to date each transportation event and consequently not every storm, they show that (i) this limitation is not restricted to OSL-RSED but an inherent problem of all established dating approaches applicable to coastal boulders; (ii) ages of initial onshore transport can give a good impression of the recurrence patterns of storms/tsunami if sufficient boulders are dated, particularly since with increasing age specific events cannot be discriminated chronologically anyway (the fact that the scenario described by the reviewer is not reflected by the small number of ages presented in this study does not mean that the principles behind it do not generally apply); and (iii) boulder movement is often not controlled exclusively by wave power, but it is typically the dominant factor. This means that using coastal boulder records for reconstructing the history of extreme wave events may be limited by some of your concerns, but since they are the best (and often only) archive available for the reconstruction of storm/tsunami impact over geological timescales, these limitations (which apply to all dating approaches, not only OSL-RSED) are widely accepted.

To sum up our reply to the general conceptual issues, it is particularly the potential of OSL-RSED that makes it a promising candidate for providing chronological information on non-limestone, quartz- and/or feldspar-bearing boulder deposits and to make use of the coarse clast record for reconstructing extreme event histories. The exposure dating has also the potential to provide depositional ages, which is preferred in comparison to dating of marine organisms prone to reworking. We consider this a chance to explore the coastal coarse clast record, and this paper shall present a step forward by evaluating and testing the potential. As we have argued before, the conceptual concerns of reviewer 2 are unsubstantiated.

**OSL-RSED data presentation:**

Comment: I find the presentation of profile data in Figs. 4, A14-16 cluttered and obscure. The mean data points with standard errors include all the information one needs to evaluate the reliability of individual data points and the overall progress of the bleaching front in a given surface. These are also the data points that are fitted to derive either the exposure age or erosion rate. So, in my view, the presentation of individual aliquots and cores in the way it is done in Figs. 4, A14-16 does not provide any useful information and impedes a proper assessment of the quality of the data.

The fits to the profile data that are used to derive the parameter values in Table 2 are not shown. Without the fits, one cannot evaluate their goodness and the reliability of the resulting parameter values.

In order to enable a clear evaluation of the data, my suggestion is to only present the mean data points with standard errors and the fits to the mean data.

Reply: Thank you for this comment. We understand the criticism of the way the OSL signal-depth data of the individual samples was presented. Our reasoning for presenting the data the way it was done in the original submission was to show the reader the entire data set he analyses is based on. We, however, realized that this may rather distract from the important information, which are the mean values and the fit of the data.

Changes in the revised version: In the revised version we followed the suggestion of reviewer 2 and adjusted Figures A14 and A15 by (i) presenting only average values for each depth, (ii) plotting the associated fit of the data to allow evaluation of its reliability, and (iii) providing the values for µ and sigmaphi_0 used for fitting each sample.

**OSL-RSED calibration:**

Comment: The data from calibration sample RAB 5-1 CAL in Fig. 5 seem to reach a plateau at _0.8 and not 1. This makes me wonder i) why this sample was normalised differently and ii) how this apparently different normalisation must have affected the calibration values derived from this sample, and hence the mean calibrated parameter values used to derive the exposure ages/erosion rates. I note that the same (mean) data presented in Fig. A18 seem to have been normalised correctly. This needs to be revised, in case the authors choose the keep this sample in a new analysis of calibration data. Please see my comment below.

345  Reply: Sorry for the confusion. There has been a mistake in the axis configuration of this sample in Figure 5. The data set used for model calibration in the original submission was, however, based on values normalized to 1.0 already. Thus, the calibration results were not affected by this issue.

350  Changes in revised version: The axis configuration in figure 5 was adjusted.

Comment: The data from calibration samples VAL 4-1 CAL 2 and RAB 5-1 CAL seem to be much more scattered than those from the other samples. Given the goodness (badness?) of the fits to such poor-quality data, I do not think that the parameter values derived from these
355  samples can be deemed as reliable. It is also intriguing that although the data from these samples are much more scattered than those from e.g. sample TEM 3-1 CAL, the relative uncertainties on sample-specific sigmaphi_0 values derived from these samples are smaller than the uncertainty on the corresponding value obtained for sample TEM 3-1 CAL.

360  Reply: It is absolutely right that these two samples are much more scattered than the others and we agree that individual values fitted using the data are not reliable. We therefore only used them in combination with the two other samples with flat surfaces to fit mutual sigmaphi_0 values.

365  Changes in the revised version: As suggested, we excluded samples VAL 4-1 CAL 2 and RAB 5-1 CAL from model calibration in the revised version.

Comment: It is argued that the sample-specific μ values have "huge uncertainties", and therefore site-specific values of μ have been derived instead as "a reasonable and necessary
370  compromise". This argument is not supported by the presented data, and is not in accordance with our understanding of μ as a physical parameter.

Firstly, the relative standard deviation (RSD) of sample-specific μ values derived from the calibration samples in Fig. 5 is ~34%, while the RSD of the corresponding sigmanphi_0 values
375  is ~210%. So, if sample-specific μ values can be dismissed because of large uncertainties and overdispersion, how can sample-specific sigmaphi_0 values, which have even greater uncertainties and are more dispersed, be acceptable and taken as a shared parameter between the calibration samples?

380  Secondly, if μ is dependent on lithology and all samples come from the same calcarenite bedrock, why not sharing μ between all the samples from all the sites? There is no evidence (or at least not presented here) that bedrock lithology varies from one site to another, so I cannot really see the logic behind sharing μ between samples from individual sites, but not between all the samples.
385
Reply: We cannot really follow the argument in this comment. We did not use sample-specific sigmaphi_0 values for calibration. We used a mutual value for all samples (otherwise we would need individual calibration samples for each targeted boulder). Thus we followed the same approach as for μ, i.e. improving the reliability of fitting by sharing the same value for several
390  samples.

While mutual sigmaphi values are, according to current knowledge, a realistic assumption for boulder surfaces from the same area and with the same surface inclination, mutual μ values indeed do not reflect the heterogeneity of rocks even from the same lithological formation (e.g.
395  Gliganic et al., 2019). This is also the case for the study site. Although the lithology is generally

similar (all calcarenite) for all boulders targeted in this study, it is not completely uniform along the entire coastline. There are slight differences in granulometry and content of bioclasts. As we explain in the original manuscript version, the best way to account for expected differences in lithology would be to use a sample-specific μ value for each sample. This is, however, impeded by fitting uncertainties, which lead to unreliable sample-specific values. We therefore have to use several samples to derive a mutual μ value. To account at least for lithological differences between the different study sites, for each of which a sufficient number of samples is available, we decided to calculate site-specific μ values in the original submission. We, however, realized that the reasoning for site-specific μ-values was not explicitly mentioned and that using a mutual μ value for all samples, as suggested by the reviewer, might indeed improve the robustness of the data (the number of samples the value is based on is much larger).

Changes in the revised version: In the revised version we followed the suggestion of the reviewer and used a mutual μ value of 1.39±0.15 for all samples. All analysis (calibration, age calculation, erosion modelling) were redone with this value. All figures and tables were updated accordingly.

Comment: The issues mentioned above make me wonder about the robustness of the calibration approach undertaken here and the reliability of the resulting parameter values. To address these issues, I would reanalyse the calibration data by i) excluding the inferior data of samples VAL 4-1 CAL 2 and RAB 5-1 CAL, and ii) sharing μ between all samples or leaving it as a free sample-specific parameter in fitting.

Reply: According to our replies above, we reanalysed the calibration data. While these modifications change the individual ages of each boulder, the overall chronological pattern of the boulders and, thus, our main conclusions are not affected.

Changes in the revised version: To reanalyse the calibration data we (1) excluded the strongly scattered samples VAL 4-1 CAL 2 and RAB 5-1; (2) started with the calculation of a mutual μ value of 1.39±0.15 mm$^{-1}$ for all samples by simultaneously fitting all samples (calibration samples and samples of unknown age) with μ as free and shared parameter and age x sigmaphi_0 (see e.g. Sohbati et al., 2015) as a single free and unshared parameter; and (3) calculated the mutual sigmaphi_0 using the calibration samples and the mutual μ (lines 325-355). All analysis based on the signal-depth data and all data resulting from them were recalculated with these new values. While the exact numbers are different, the main conclusions do not change.

**Erosion rate modelling:**

Comment: The authors have followed a numerical approach (not "analytical" as is mentioned in line 333) to model the OSL erosion rates. But, the OSL erosion rate equation has an exact analytical solution that is already published (see Sohbati et al., 2018). So, there is no need and no scientific justification for making guesses at the solution numerically as is done here. The parameter values derived from the calibration samples can simply be inserted in the erosion rate equation and fitted to the profiles to give erosion rates.

Reply: The approach of Lehmann et al. (2019) that we applied to our samples is indeed a numerical approach. We, however, completely disagree that the application of a numerical approach lacks scientific justification while an analytical approach exists. We are aware that an analytical solution for the quantification of erosion from OSL rock surface data was already

presented by Sohbati et al. (2018). The numerical approach of Lehmann et al. (2019) that is used in this study was later published in Earth Surface Dynamics, acknowledging the analytical approach but providing an alternative solution for the erosion problem. Both approaches have their advantages and there is no approach that is absolutely superior compared to the other. The analytical solution of Sohbati et al. (2018) might be more elegant and faster, but the numerical approach chosen in this study (which is not guessing, but inferring results from our data) is able to resolve the problem in time and provides a quantification of misfits and thus uncertainties on the results.

Changes in revised version: We refer to the Lehmann et al. (2019) model as a numerical approach in the revised version of the manuscript (line 389).

**Minor comments:**

Comment, Line 17: I suggest "wave-driven" instead of "wave-emplaced". The boulders cannot be "emplaced" by waves and "transported" at the same time.

Changes in the revised version: The wording was changed to "wave-driven".

Comment, Lines 48-49: "...these approaches are restricted to certain boulder lithologies and time scales.". So is OSL RSED; it is largely restricted to lithologies that "contain quartz and/or feldspar" and to timescales of "decades, centuries up to a few millennia" as is mentioned later in lines 61-62.
Reply: Thank you for this comment. We realized that we have to be more specific here. Palaeomagnetic dating still suffers from a number of intrinsic methodological limitations, and cosmogenic nuclide dating typically cannot provide sufficient resolution on Late Holocene time scales and is, therefore, of limited benefit for the vast majority of coastal boulders.

Changes in the revised version: We added a sentence explaining the limitations of the two dating approaches more specifically in the revised version of the manuscript (lines 49-53).

Comment, Line 63: Does the statement "...to reconstruct...tsunami frequency patterns..." imply that the tsunami events are expected to follow some sort of temporal/spatial patterns?

Reply: Yes, tsunamis typically show temporal patterns if they are generated by earthquakes. Since the 1755 Lisbon tsunami was triggered by an offshore earthquake, it is not unlikely that potential predecessors follow a certain temporal pattern that is controlled by the accumulation of seismic strain.

Comment, Lines 71-72: Consider to change "...erosion of post-transport exposed boulder surfaces..." to "erosion of boulder surfaces exposed after transportation" or something like that.

Changes in the revised version: The wording was changed accordingly.

Comment, Line 79: Add "buried" before "sediment".

Changes in the revised version: Will be changed as suggested.

Comment, Line 94: What Fig. 1 is actually showing is a boulder that is detached from a wave-cut platform and overturned by waves. There is no "transportation" involved in the depicted scenario.

Reply: The relocation of the boulder from the cliff edge to the supratidal coastal platform in an overturning movement clearly involves transportation. In Figure 1 the process of overturning during transport is illustrated by showing two successive stages of boulder movement.

Changes in the revised version: We nevertheless changed the wording in the revised version to better express the fact that we always date the cliff detachment of overturned boulders and not potential transport events following afterward, which typically take place as a sliding movement for plate-shaped boulders as selected in this study (lines 105-120).

Comment, Line 147: I cannot see how 2-3 m-high spring tides can reach and exceed the 5-m high first ridge (as is mentioned in line 154) to flood Oulja.

Reply: While the first calcarenite ridge shows average heights of about 5 m above sea level, this barrier occasionally shows sections with lower elevations or can even be breached at river mouths. This is where water can enter the depression of the Oulja during high tides.

Comment, Lines 189-196: The preheat temperature should also be mentioned somewhere in these lines as Table A2 is in the Appendix.

Changes in the revised version: In the revised version, we now mention the preheat condition, i.e. 220 °C for 10 s (line 221).

Comment, Line 191: The stimulation time in Table A2 is 150 s and not 160 s.

Reply: Sorry for this mistake. This should be 160 s as stated in the main text.

Changes in the revised version: Table A2 was updated accordingly.

Comment, Lines 197-208: I suppose the dose recovery and preheat plateau tests described in this paragraph were carried out to guide decision on the most suitable measurement protocol. In that case, this paragraph must precede the previous paragraph in which the actual measurement protocol is explained.

Changes in the revised version: We changed the order of arguments to clarify that these experiments were used as a basis for final protocol selection.

Comment, Line 207: The "burial ages" suddenly appear here. So far, only OSL RSED is discussed. It is also mentioned (in lines 104-105) that the buried sides of the boulders are inaccessible and "not tried in this study". So, speaking of burial ages here is confusing to me. In fact, it is first 60 lines further down in the text (line 267) that a careful reader may find out that what here is referred to as burial age, is actually the rock formation age, calculated by dating quartz extracts from deep layers within the boulders that have never seen light after rock formation. These should not be confused by boulder surface burial ages.

Reply: Sorry for the confusion. While the ages indeed reflect the timing of sand grain burial during ridge formation, we agree that the term "burial age" may be ambiguous in this study.

Changes in the revised version: To differentiate rock surface burial ages (which were not determined in this study) from conventional OSL dating of the sandstone formation (which we refer to here), we replaced "burial ages" by "ages for sandstone formation" in the revised version.

Comment, Line 212: I find the use of the term "background level" inappropriate here. Background level in OSL dating is commonly referred to while discussing the stimulation curves. I suggest "plateau" instead.

Reply: We absolutely agree that the term "background" may be misleading in this context.

Changes in the revised version: The wording will be changed as suggested.

Comment, Lines 214-215: This sounds to be a subjective and qualitative approach towards removing the outliers, while there are various quantitative methods to identify them. One common approach that could also be used here is to remove those data points that are different than the mean by three standard deviations.

Reply: Thank you for the suggestion. Our approach was indeed somehow subjective.

Changes in the revised version: We revised our rejection criteria as follows: (i) Entire cores were excluded, if they did not show any signs of bleaching with depth, while all other cores from the same sample did. (ii) All other data points were classified as outliers according to a deviation from the mean of more than 2 standard deviations (lines 245-247). We use this new data set for all analysis in the revised version. The data, however, did not change significantly.

Comment, Line 229: Not sure what is meant by "comparable preconditions for sunlight exposure". If the scenario is as simple as shown in Fig. 1, then all the boulders must have experienced comparable conditions (i.e. detachment and overturn). But if they are likely to have been reworked (i.e. moved and turned over multiple times) then it is very difficult to imagine how they could have had comparable exposure conditions.

Reply: The meaning of this term is explained in the second part of this sentence. While all boulders used for this study have been overturned only once (see reply to main comment: the platy boulders used in this study were overturned when detached from the cliff, but moved by sliding only or not at all afterwards), sunlight exposure may also be different due to differential shielding after deposition or due to different exposure angles.

Changes in the revised version: We changed "preconditions" to "conditions", since this term seems more appropriate.

Comment, Line 252: How about "target" instead of "dated"?

Changes in the revised version: Changed as suggested in the revised version.

Comment, Lines 253-256: It is difficult for me to judge this inference by the way the data are presented in Fig. A12. The pure quartz BSL, K-rich feldspar IRSL and polymineral post-IRSL-BSL signals must be normalised and shown on the same graph to enable a direct comparison.

Reply: We do not agree with this opinion. What we want to document is: (1) Post-IRSL-BSL signals of polymineralic aliquots are significantly stronger than the IRSL signals measured on

the same polymineralic aliquots. This is documented in Fig. A12a, where normalized values of both signals are compared in the same plot (as asked for by the reviewer). (2) The post-IRSL-BSL signals of polymineralic aliquots are dominated by a quartz signal with only minor influence of feldspar signals. This is documented in Fig. A12d, which shows that the IRSL stimulation used in our protocol reduces the potassium feldspar signal to 60% of its initial value (more details are given in the caption of Fig. A12).

Comment, Lines 274-275: I assume that calibration was carried out before fitting the actual data? Please present the steps in data analysis in the logical order.

Reply: We used this sentence as an introduction to the explanation, why calibration is necessary.

Changes in revised version: To avoid confusion, we changed the wording to "To estimate boulder ages with OSL-RSED, measured post-IRSL-BSL signal-depth data must be fitted with the bleaching model described in Equation (1)".

Comment, Line 279: Sohbati et al. (2011) is the correct reference.

Changes in revised version: The reference was changed accordingly.

Comment, Lines 293-295: This is an interesting observation that the calibration sample TEM 3-1 CAL that is collected from an inclined surface yields a sigmanphi_0 value that is ~3 orders of magnitude larger than the corresponding values estimated for the horizontal surfaces. If this conclusion still stands after data reanalysis (see my comments above), it would be useful to report the tilt angle of the surface. At the moment, there is no data on the dependence of sigmanphi_0 on the incident angle of solar radiation in the literature.

Reply: After reanalysing the data by excluding the two calibration samples with poorly defined bleaching fronts, there is still a significant difference of one order of magnitude between the horizontal calibration samples and the inclined calibration sample (the angle of the surface is already reported in Table 1 with ~25°). We agree that such an observation has not been reported and would be worth a more detailed investigation. We are, however, aware that our assumption is only based on a single sample (and a total set of 5 calibration samples even without excluding RAB 5-1 CAL and VAL 4-1 CAL2). It obviously needs a larger dataset and more controlled conditions (e.g. in a bleaching experiment) to evaluate the assumed relationship between inclination of the surface and sigmaphi_0.

Comment, Line 307: What is meant by "inadequate" here?

Changes in revised version: We changed "inadequate" to "incorrect".

Comment, Line 333: The approach of Lehmann et al. (2019) is numerical not analytical.

Changes in revised version: We changed "analytical" to "numerical" in the revised version.

Comment, Line 356: "observed" instead of "achieved"?

Changes in revised version: The wording was changed accordingly.

Comment, Line 365: It seems unlikely to me that "mineralogy-induced dose rate differences" can result in the observed scatter in data from such samples. Hot minerals such as zircon and

K-rich feldspars are rare, if not non-existent, in calcarenite. Meyer et al. (2018) have attributed similar scatters in their data to the presence of opaque minerals and iron hydroxides, which strongly impede the penetration of light with depth. In the absence of any independent evidence, this seems more reasonable to me as an explanation here.

Reply: We agree that this argument will definitely not explain most of the observed scatter. While it may add to the observed scatter of signals (that is why we included the argument originally), it is likely of very minor importance and might involuntarily make the discussion more complicated than necessary.

Changes in revised version: We decided to abstain from using this argument in the revised version of the manuscript.

Comment, Lines 366-368: I am not sure I follow. How can the aliquot-to-aliquot variation in feldspar content can give rise to additional scatter in profile data? Does it mean that test dose is not adequately correcting for this possible variation? Why not? What is the evidence?

Reply: We do not have direct evidence for this argument. We however know that the IRSL stimulation of our post-IRSL-BSL protocol is removing most of the feldspar signal, but not all of it. Although we assume that the feldspar contribution is insignificant based on our test measurements, it must be expected that the contribution of feldspar signals to the post-IRSL-BSL signal will be slightly different for polymineralic aliquots with different percentages of feldspar.

Changes in revised version: Since this potential source of scatter will again explain (if at all) only a very minor part of the observed scatter, we again decided to abstain from using this argument in the revised version of the manuscript and focus on the most plausible arguments (lines 420-435).

Comment, Lines 375-378: While the interpretation that age underestimation could have been caused by unreliable sigmaphi values and erosion of the boulder surfaces may be right, it would nevertheless be interesting to see what erosion rates one would get by applying the erosion rate model to samples that do not seem to suffer from age underestimation. The erosion rate of such samples must be negligible compared to the erosion rates of the samples showing age underestimation. This should provide a good basis for your interpretation.

Reply: The point developed by reviewer 2 is fair and was tackled during revision of the manuscript. The erosion rates provided by the model are indeed negligible (i.e. < 0.01 mm/year). This is, however, not surprising given the fact that the depth profiles of these samples can be explained without any erosion.

Comment, Line 377: Does "inadequate" mean "unreliable" here?

Changes in revised version: We replaced "inadequate" by "unreliable".

Comment, Line 388: What is meant by "environmental factors beyond the exposure time"?

Reply: The factors refereed to here, i.e. post-transport erosion and occasional shielding of the post-transport surface by e.g. water, are explained in the following sections of the manuscript.

Changes in revised version: We changed the wording to "factors different than exposure time" to avoid any confusion.

Comment, Lines 418-419: It may be worth mentioning here that, in retrospect, IRSL signals were likely to work better than the post-IRSL-BSL signals for these samples.

Changes in revised version: We added a short reference regarding the potential benefits of IRSL signals for some of our samples: "While IRSL signals were not used in this study due to insufficiently bright signals for most samples, in retrospect their use might be advantageous to post-IRSL-BSL signals at least for some of the investigated samples" (lines 499-501).

Comment, Line 422: What is considered as "insufficiently bright signals"? If the post-IRSL-BSL signals shown in Fig. A12 are typical for these samples, they are all well above background by more than 3.

Reply: The term "insufficiently bright" refers to the IRSL signals of polymineralic samples. Those shown in Figure A12a are representative for the samples of the different sites and hardly distinguishable from the background (signal <3 times background for most aliquots).

Comment, Line 431-434: 1) The ages obtained from eroding surfaces are "apparent" surface exposure ages. The fact that they underestimate the expected ages, does not mean that they are inaccurate. They may be accurate, but they simply do not reflect the age of the event of interest. 2) There is no scientific basis to support this general statement that the ages from inclined surfaces are inaccurate. Surfaces can be dated regardless of their orientation provided that suitable calibration samples are available.

Reply: We agree with and appreciate these arguments. What we want to express is that apparent ages do not agree with age control due to erosion or unreliable calibration samples.

Changes in revised version: We changed the phrasing of the section to better reflect this argumentation in the revised version (lines 514-515).

[revised manuscript text omitted]

**Fig. 8: Relative chronology of boulder transport. a)** Exposure ages of all boulders that do not clearly underestimate the control ages presented as KDE plot (dotted error bars with consideration of $\mu$ and $\overline{\sigma\varphi_0}$ uncertainties). The numbers in squares refer to the taphonomy classes described in the text and the caption of Table 1. Inset: Correlation between boulder mass and OSL rock surface exposure ages. **b)** Photographs documenting the taphonomy of boulders with different OSL rock surface exposure ages. Each photo is correlated with a KDE peak in a) and  associated boulder mass by dashed lines.

| Boulder | Depositional setting | | Group | Boulder dimension | | | | Sampled surface | | Sample | |
|---|---|---|---|---|---|---|---|---|---|---|---|
| | Lat/Long (°) | Position | | Shape | Vol. (m³) | Density (g/m³) | Mass (t) | Orientation | Tapho-nomy | Dating | Calibration |
| RAB 1 | 34.007964 -6.869395 | Supratidal platform 2.7 m asl, 40 m from cliff | Single | Irregular | 15.3 | 2.45 | 37.5 | 0° 25° SE | 3 | RAB 1-1 RAB 1-2 | - |
| RAB 5 | 33.999778 -6.877759 | Intertidal platform (surface above tides) | Single | Cubic | 9.8 | 2.4 | 23.5 | 0° | 2 | RAB 5-1 | RAB 5-1 CAL |
| HAR 1 | 33.953904 -6.927609 | Oulja 2 m asl, 35 m from cliff | Single | Platy | 8.2 | 2.3 | 18.9 | 0° | 4 | HAR 1-1 | HAR 1-1 CAL** |
| HAR 2 | 33.953525 -6.928669 | Top of younger ridge 4.5 m asl, 16 m from cliff | Imbricated | Platy | 3.6 | 2.4 | 8.6 | 25° NW | 4 | HAR 2-1 | - |
| HAR 3* | 33.954182 -6.927901 | Niche in active cliff (intertidal) | - | Platy | - | - | - | 0° | 3 | HAR 3-1 | - |
| TEM 2 | 33.927066 -6.961859 | Top of younger ridge 4.2 m asl, 35 m from cliff | Single | Platy | 8.3 | 2.4 | 19.9 | 0° | 2 | TEM 2-1 | - |
| TEM 3 | 33.926842 -6.961915 | Top of younger ridge 4.5 m asl, 38 m from cliff | Single | Irregular | 0.3 | 2.4 | 0.7 | 25° W | 5 | TEM 3-1 | TEM 3-1 CAL |
| TEM 4* | 33.927949 -6.960674 | Niche in active cliff (supratidal) | - | Platy | - | - | - | 25° W | 5 | TEM 4-1 | - |
| VAL 1 | 33.909435 -6.989803 | Intertidal platform (surface above tides) | Single | Cubic | 26.2 | 2.25 | 59.0 | 10° SE | 1 | VAL 1-1 VAL 1-2 | - |
| VAL 4 | 33.907733 -6.991505 | Top of younger ridge 3 m asl, 25 m from cliff | Imbricated | Platy | 7.3 | 2.2 | 16.1 | 0° | 3 | VAL 4-1 | VAL 4-1 CAL1 VAL 4-1 CAL2** |
| VAL 6 | 33.906084 -6.993316 | Oulja 2.5 m asl, 80 m from cliff | Single | Platy | 8.9 | 2.3 | 20.5 | 0° | 4 | VAL 6-1 | - |

**Tab. 1: Characteristics of dated boulders. Lat/Long = Latitude/Longitude, \* = niche at coastal cliff, \*\* = calibration sample on nearby roof top, taphonomy classes 1 to 5 with 1 – post-transport rock pools, 2 – rough post-transport surface covered with lichens/algae, 3 – smooth post-transport surface with scarce lichen/algae cover, 4 – smooth post-transport surface without/hardly any lichens/algae and fresh fractures, and 5 – fresh post-transport surfaces and fractures.**

[revised manuscript text omitted]